# Turn-on luminescence from molecular rotor realignment in metal-organic framework thin films

Jan C. Fischer[1,8], Tong Zhou[2,3,8], Philipp Sievers[4], Nils W. Rosemann [1,5], Elizabeth Coetsee[6], Dmitry Busko[1], Yang Li [1], Honghan Ji[2,3], Diethelm Johannsmann [4] ✉, Lingju Guo[2,3] ✉, Pengfei Duan [2,3], Bryce S. Richards [1,5], Ian A. Howard [1,5,7] & Tonghan Zhao [1] ✉

Sub-unit motion within metal-organic frameworks (MOFs) offers unique opportunities for nanoscale sensing. However, achieving controlled partial rotation of bulky linkers remains a significant challenge. In this study, a 50-fold luminescence enhancement is observed from a MOF thin film when intra-pore solvent flow orients the linker chromophores. These MOF thin films can be prepared via a facile drop-casting method on various substrates. The MOFs structure consists of zinc-coordinated layers containing rotatable chromophores, separated by pillar molecules. Grazing-incidence wide-angle X-ray scattering analysis confirms the formation of highly oriented films. The deposition of a volatile organic compound, such as ethanol, triggers a significant enhancement in luminescence as the solvent nears complete evaporation. Photophysical characterization and quartz crystal microbalance measurements reveal that this phenomenon is driven by internal stress on the MOF's pore level generated during the final stages of evaporation. This stress can result in a realignment of the MOF chromophores at the molecular scale. Consequently, this dynamic turn-on luminescence behavior establishes a foundation for nanoscale platforms capable of indicating solvent volatilization in real time.

Controlled molecular motion within crystalline solids attracts significant attention for its potential to create molecular machines exhibiting diverse functionalities. Nanoscale dynamic processes, including rotation, shuttling, and switching, are pivotal for engineering materials with novel macroscopic properties[1–7]. However, solid-state molecular motion is considerably challenged by the spatial confinement, intrinsic rigidity, and intermolecular interactions characteristic of crystal lattices[8,9]. Metal-organic frameworks (MOFs), constructed from organic linkers and metal ions, provide ideal platforms for the precise spatial arrangement of mobile molecular components. The highly porous cavities within MOFs generate significant internal free volume, which accommodates translational and rotational degrees of freedom. Additionally, the porosity of MOFs permits the diffusion of guest molecules, thereby facilitating access for external chemical species to

[1]Institute of Microstructure Technology, Karlsruhe Institute of Technology, Hermann-von-Helmholtz-Platz 1, Eggenstein-Leopoldshafen, Germany. [2]National Center for Nanoscience and Technology, No. 11 ZhongGuanCun BeiYiTiao, Beijing, People's Republic of China. [3]University of Chinese Academy of Sciences, Beijing, People's Republic of China. [4]Department of Physics, University of the Free State, Bloemfontein, South Africa. [5]Light Technology Institute, Karlsruhe Institute of Technology, Engesserstrasse 13, Karlsruhe, Germany. [6]Institute of Physical Chemistry, Clausthal University of Technology, Arnold-Sommerfeld-Str. 4, Clausthal-Zellerfeld, Germany. [7]Present address: Carl Zeiss AG–Innovation Hub Karlsruhe, Hermann-von-Helmholtz-Platz 6, Eggenstein-Leopoldshafen, Germany. [8]These authors contributed equally: Jan C. Fischer, Tong Zhou. ✉e-mail: johannsmann@pc.tu-clausthal.de; guolj@nanoctr.cn; tonghan.zhao@kit.edu

interact with and modulate the internal molecular dynamics[10–13]. MOFs incorporating molecular rotors (a molecular component that usually consists of a stator and a rotator, where the latter can show rotary motion) are of particular interest due to their potential to combine low rotational barriers with rotor-rotor interactions within a three-dimensional solid. Within these integrated architectures, molecular rotors are typically incorporated as linkers, wherein the stator component coordinates with metal ions to construct the MOF. These materials exhibit internal dynamics and respond to environmental stimuli, including changes in pressure, temperature, viscosity, and guest molecules[14]. This responsiveness highlights the potential of these dynamic systems for applications in gas adsorption, molecular recognition, drug delivery, and water desalination[15].

Various molecular rotors in MOFs have been investigated, featuring rotational axles formed by hybridized orbital bonds such as $sp^3$–$sp^2$, $sp$–$sp^2$, and $sp^2$–$sp^2$ [16–19]. Among these, luminescent molecular rotors represent a promising class of linkers, characterized by their ability to respond to external stimuli through distinct changes in photoluminescence (PL) properties[20,21]. Particular attention has been paid to rotor ligands that exhibit aggregation-induced emission. For instance, Zhao et al. reported a series of tetraphenylethylene-based MOFs that demonstrate turn-on fluorescence in response to volatile organic compounds, temperature variations, and viscosity changes[22,23]. Recently, Zhu et al. synthesized a triphenylamine-based lanthanide MOF wherein the distortion of molecular rotors is regulated by temperature, thereby inducing multicolor luminescence switching[24]. In these reported systems, phenyl rings function as the rotators, a mechanism that requires only a relatively small free volume. However, incorporating bulkier units, such as anthracene, significantly increases the rotational barrier. This steric hindrance can lead to restricted partial rotation or even stabilize the molecule into a stationary conformation[25–27]. To address this challenge, an external stimulus is required to trigger the rotation of the molecular rotors. Subsequently, upon the withdrawal of this stimulus, the high rotational barrier drives the MOF rotors to revert to the initial conformation. This principle provides an opportunity for the development of reversible optical sensing functionalities. For these applications, the deposition of MOFs as thin films is a particularly promising strategy[28–31]. A critical determinant of performance in these MOF thin films (MTFs) is their preferred crystallite orientation, which modulates host-guest interactions and optical properties[32,33].

This work demonstrates <001>-oriented MOF thin films where the final stage of solvent evaporation triggers the reversible partial rotation of a bulky anthracene-based linker (Fig. 1a, b). Molecular rotation induces long-range alignment of the anthracene moieties and activates their photoluminescence (PL). Upon evaporation to dryness, the rotors revert to their relaxed state, and PL is quenched. Specifically, the MTFs are composed of a layer linker (9,10-anthracene dicarboxylic acid, ADC), a pillar linker (1,4-diazabicyclo[2.2.2]octane, DABCO or 4,4′-bipyridine, BPy), and zinc (Zn) ions, manifesting stacked Zn-ADC paddle wheel networks parallel to the substrate. To fabricate pillared-layer MTFs with preferential orientation, a straightforward drop-casting technique is employed (Fig. 1a). Typically, highly oriented pillared-layer MTFs are produced via layer-by-layer deposition, a process involving the sequential exposure of a substrate to metal and linker precursor solutions[34–36]. While these conventional techniques afford control over the crystal growth of ADC-based MTFs, they are often time-consuming. Furthermore, they generally require surface functionalization with a self-assembled monolayer to support and direct oriented growth[37,38]. In contrast, the drop-casting method presents a simple and effective alternative, yielding a 3D pillared-layer MOF with the desirable <001>-orientation. To the best of our knowledge, this approach has not been previously reported. Its simplicity makes it highly attractive, offering significant potential for facilitating device integration. The resulting structure and orientation

are validated using grazing-incidence wide-angle X-ray scattering (GIWAXS) analysis. In contrast to phenyl-based rotors, the anthracene moiety is typically regarded as a sterically hindered rotor. Consequently, within the dense Zn-ADC-DABCO framework, its PL is quenched[39]. However, the application of ethanol to the MTFs triggers a dramatic change. Intriguingly, as the solvent approaches complete evaporation, the MTFs exhibit a sharp PL turn-on phenomenon. This response yields a 50-fold enhancement in intensity relative to the initial dry state. Photophysical and quartz crystal microbalance (QCM) investigations suggest that such enhancement arises from a transient realignment of MOF linkers, a process probably driven by internal stress generated during the final stage of solvent evaporation. This work presents a proof-of-concept demonstration of the utility of these MOFs as platforms for evaporation indicating, offering potential for the development of advanced sensing devices.

## Results

### Oriented metal-organic framework thin films fabricated by drop-casting

Pillared-layer structures represent a useful topology for investigating solid-state dynamics, offering substantial free volume for molecular rotary motion. Here, we introduce a facile drop-casting method for the rapid and reproducible synthesis of highly oriented pillared-layer MTF. The method involves depositing a freshly prepared ethanolic solution of precursors onto a < 100 > silicon (Si) substrate maintained at 50 °C to evaporate solvent (see Methods). After several times of drop-casting, a Zn-ADC-DABCO MTF forms on the substrate. Scanning electron microscopy (SEM) reveals that the thin film consists of dense layers of intergrown plate-like crystallites (Fig. 1c). The chemical composition of the MOF was characterized by Fourier-transform infrared spectroscopy (FT-IR) and X-ray photoelectron spectroscopy (XPS). The FT-IR spectra of Zn-ADC-DABCO and the ADC linker are presented in Supplementary Fig. 1. For the ADC linker, absorption bands in the $2500–3100\ cm^{-1}$ region are assigned to the carboxylic O–H stretching vibrations. A characteristic absorption band at $1677\ cm^{-1}$ corresponds to the C=O stretching vibration. In the spectrum of the Zn-ADC-DABCO MTF, the broad carboxylic O–H stretching vibrations ($2500–3000\ cm^{-1}$) disappear. Concurrently, the C=O stretching vibration splits into two new bands at 1615 and $1560\ cm^{-1}$. These spectral changes indicate the formation of Zn−O bonds. XPS measurements confirm the presence of Zn, C, O, and N within the MTF (Supplementary Fig. 2). Furthermore, analysis of the high-resolution O $1s$ and N $1s$ XPS spectra reveals peak-fitting results reveal the coordination of these elements to zinc. The structure and preferential orientation of the thin film are confirmed using GIWAXS (Fig. 1d). Excellent agreement is found between the experimentally observed diffraction peak positions and those simulated from a Zn-ADC-DABCO bulk crystal model in the <001>-orientation[40]. The preferential <001>-orientation is evident from the distinct arrangement of the diffraction peak along the $q_z$ direction (substrate-orthogonal periodicity) and the {110} and {020} peaks along the $q_r$ direction (substrate-parallel periodicity). In addition, the out-of-plane X-ray diffraction (PXRD) spectrum of Zn-ADC-DABCO MTF has a good agreement with the simulated diffractogram with preferred orientation along the [001] direction (Supplementary Fig. 3), indicating the film grew along the [001] direction perpendicular to the substrate surface. This alignment confirms the structure of horizontally aligned sheets of Zn-ADC paddlewheels interconnected by vertically oriented DABCO pillar molecules (Fig. 1a).

To optimize MOF quality, the radial intensity distribution of the {112} diffraction peak was extracted as a function of the azimuthal angle χ (Fig. 1d). The degree of crystallite orientation was then evaluated by assessing the azimuthal full width at half maximum (FWHM) of this {112} diffraction peak while varying synthesis parameters, including precursor concentration, substrate temperature, and drop

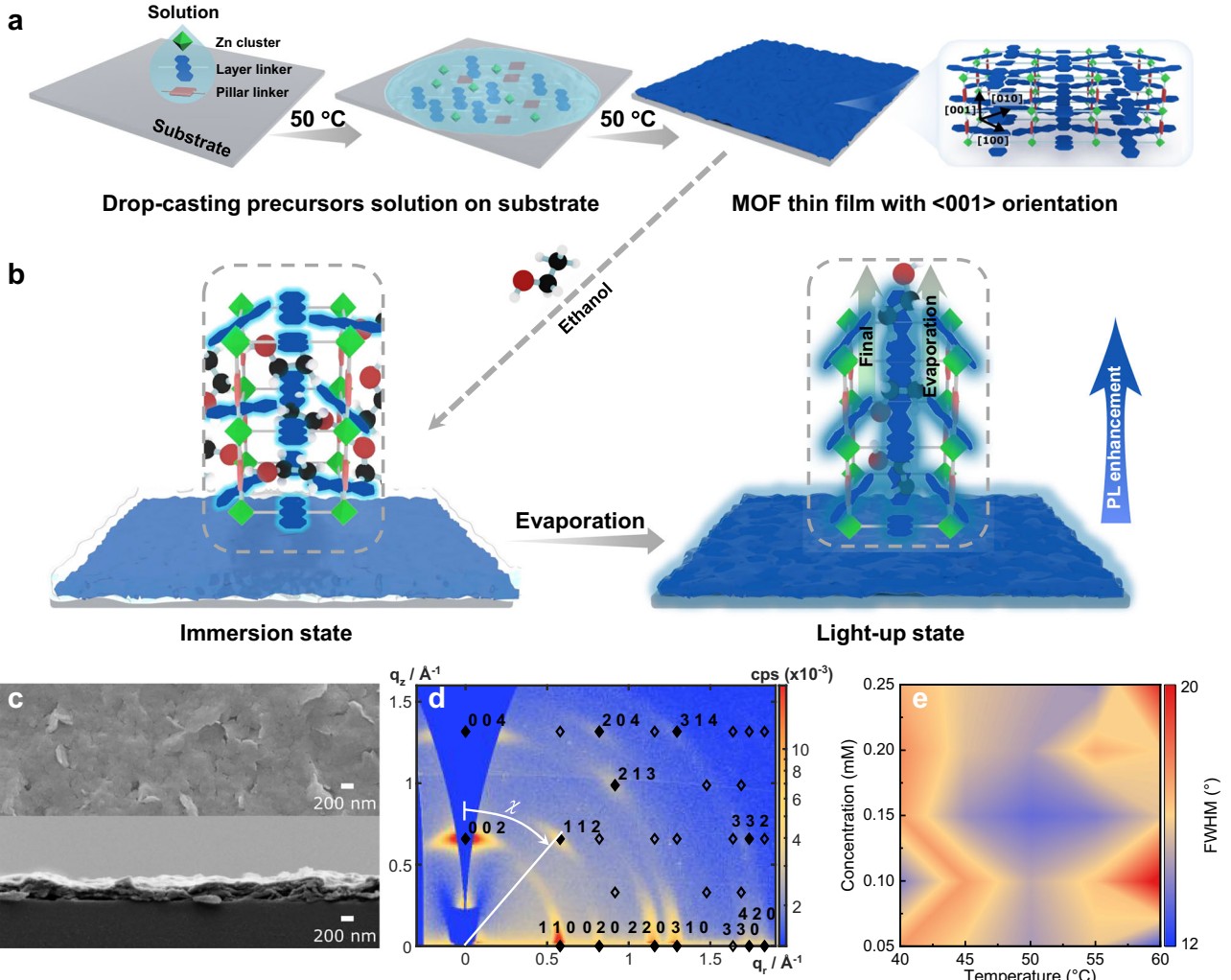

**Fig. 1 | Drop-casting of Zn-ADC-DABCO layer-pillar MOF thin films on a hot substrate. a** Ethanol growth solution, containing Zn precursor, ADC linker and DABCO pillar, drop-cast on a heated substrate, results in MTF growth with preferentially <001>-oriented MOF structure. **b** Ethanol evaporation induces significant PL enhancement of MTF. The Zn-ADC-DABCO MTF is shown in the immersion state and light-up state. In the initial dry state, the MOF pores are empty and ADC linkers in their relaxed configuration as shown in (**a**). When ethanol is added, the solvent molecules diffuse into the MOF cavities. In the light-up state, the final evaporation of ethanol generates local stress, deforming the MTF on a molecular scale. Ethanol molecules leave the MOF pores, dragging the ADC rotors into a more vertical alignment, which is accompanied by significant PL enhancement. **c** SEM images of MTF on a Si substrate (top: surface view, bottom: cross section) showing plate-like crystallites of the order of 200 nm forming a closed layer. **d** GIWAXS diffractogram confirming the MTF preferential <001>-orientation determined by an excellent match with simulated diffraction peak positions (diamonds). Laue indices are indicated for the simulated peaks that coincide with experimentally observed maxima (filled diamonds). $\chi$ is the azimuthal angle. **e** Synthesis parameter screening. The azimuthal FWHM of the {112} GIWAXS peak reflects the average deviation of crystallite angles from the perfect <001>-orientation and, therefore, provides a measure for the quality of orientation. An optimum is indicated at 0.15 mM and 50 °C. Source data are provided as a Source Data file.

volume (see Supplementary Note 1 for details). Optimal crystallite orientation, characterized by the smallest FWHM (indicating minimal dispersion in sheet alignment), was achieved using a precursor concentration of 0.15 mM and a substrate temperature of 50 °C (Fig. 1e and Supplementary Figs. 4−6). For this optimized sample, the oriented fraction was estimated to be 73% via Gaussian fitting of the peak (Supplementary Fig. 7). The stability of the Zn-ADC-DABCO film was evaluated by PXRD. As presented in Supplementary Fig. 8, the PXRD patterns of the film were collected after exposure to ambient air and immersion in ethanol for five days. The patterns confirm that the crystalline structure and orientation are retained after these treatments, indicating good stability of the MOF. Subsequent investigations were therefore conducted under these optimized conditions. Oriented thin films with good homogeneity across the entire coated area were achieved, as demonstrated in Supplementary Fig. 9. Additionally, these oriented films can be successfully deposited on various surfaces

beyond silicon, including quartz, ZnO@Si, TiO₂@Si, Gd₂O₃@Si, and Ag@Si (Supplementary Fig. 10). Similar results were obtained for an isoreticular MOF system when DABCO was replaced as the pillar molecule with the slightly longer 4,4′-bipyridine (BPy, Supplementary Fig. 11). To elucidate the formation mechanism of the oriented MOF, GIWAXS of MTF prepared from pre-crystallized solutions, crystallite thickness of MTF made with various solution volumes, and XPS of the MTF-loaded Si substrate interface were studied. As shown in Supplementary Fig. 12, an increased degree of pre-crystallization results in poor MTF orientation. This suggests that the sedimentation of preformed crystal grains is not the governing mechanism. XPS analysis of the substrate interface reveals a weak peak corresponding to Si−N bonds (Supplementary Fig. 13), indicating an interaction between the Si surface and the DABCO linker. However, the observation of a constant crystallite size, regardless of increasing solution volumes (Supplementary Fig. 14), rules out a stepwise epitaxial surface growth

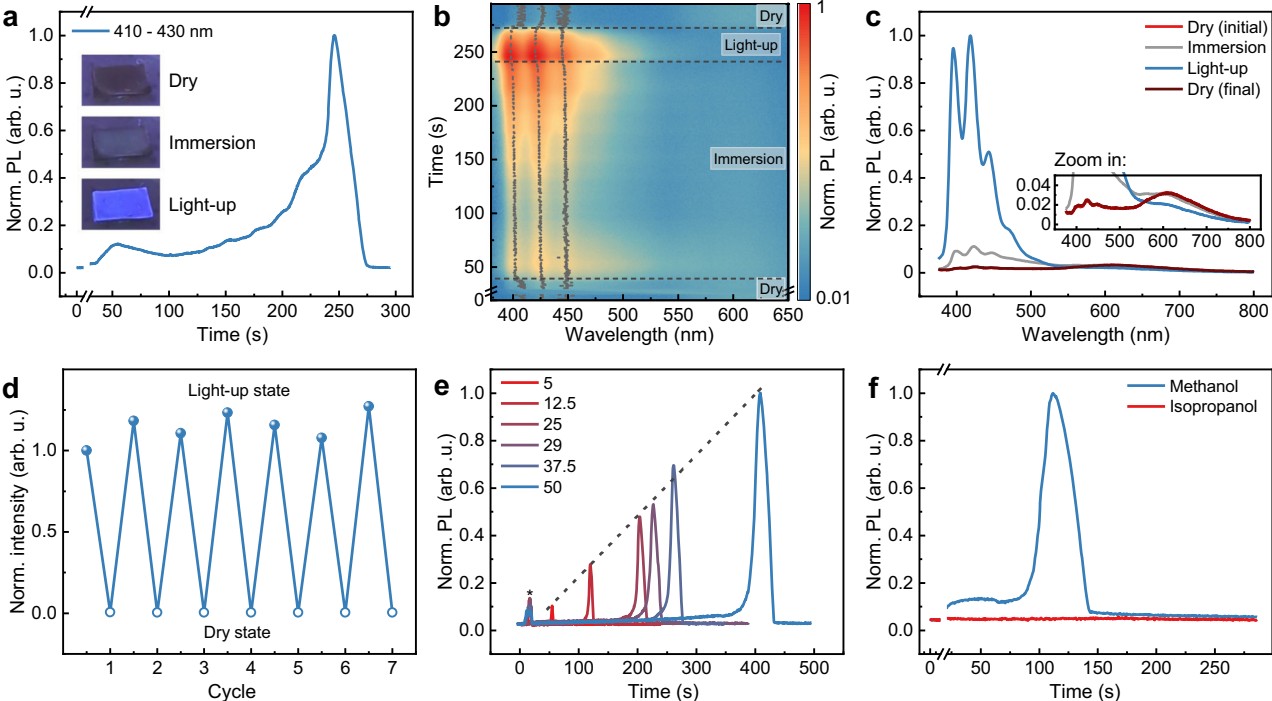

**Fig. 2 | Luminescence turn-on effect in drop-cast Zn-ADC-DABCO thin film during solvent evaporation. a** PL intensity (averaged from 410 to 430 nm) over time. The time region when the solvent was applied has been clipped, where an artefact was created by the glowing pipette tip, not related to the MOF emission. Given in insets are photos taken of a sample in the three emission phases. These phases are "dry" (initial and final state), "immersion" (thin film solvent immersion on the substrate), and "light-up" (end phase of evaporation as solvent molecules leave the MOF pores). **b** Spectral evolution of PL over time with the normalized PL intensity by a color-coded logarithmic scale. The three main PL peak positions in each spectrum are indicated by gray dots. The different evaporation phases are labeled by horizontal dashed lines. **c** Time-integrated average spectra of different evaporation states. **d** PL intensity during light-up state and dry state over several cycles of ethanol evaporation in direct succession. **e** PL intensity during evaporation of different ethanol droplet volumes (μL/cm²). The maximum PL scales linearly with the solvent volume applied (linear fit of the peak positions indicated by dashed line), while the total solvent evaporation time is, evidently, linearly increased. The region labeled by * directly is an artefact from the droplet application. **f** PL intensity measured during methanol and isopropanol evaporation. There is no luminescence turn-on for isopropanol. All graphs in **d**, **e** show the averaged PL intensity from 410 to 430 nm. Source data are provided as a Source Data file.

mechanism. Based on these results, the oriented growth of the MOF, particularly at the droplet periphery, is likely governed by a combination of two mechanisms (discussed further in Supplementary Fig. 15): (1) surface-initiated crystallization, mediated by a functional layer of pillar molecules on the substrate; (2) formation of nanoplatelets in solution, which subsequently adopt a lying-flat morphology on the substrate, driven by surface energy minimization and horizontal capillary forces during ethanol evaporation[41,42]. A key factor in this formation process appears to be the protection of the paddlewheel-pillar bond from hydrolysis by the bulky ADC rotors[43–45], as synthesis attempts with less bulky linkers did not yield films of comparable quality (Supplementary Fig. 16).

### Luminescence turn-on effect

Given that the ADC linker in ethanol exhibits bright PL with a quantum yield (PLQY) of 45.5% (Supplementary Table 1), the emission properties of the corresponding Zn-ADC-DABCO thin films were investigated. In contrast, the thin films display significantly weaker PL, with a PLQY of only 0.667%. This pronounced discrepancy in PLQY has previously been attributed to self-quenching effects arising from dense chromophore packing within the MOF structure[39]. Correspondingly, the temperature-dependent fluorescence lifetime spectra revealed a low activation energy for non-radiative transitions (Supplementary Fig. 17), indicating that non-radiative processes have a significant influence on PLQY at room temperature. To explore potential activation of the Zn-ADC-DABCO emission, various solvents (n-hexane, cyclohexane, toluene, ethanol, and acetone) were applied to fully cover the thin film surface. A strong effect was observed exclusively with ethanol (applied

at 25 μL/cm²). Monitoring the emission during solvent evaporation reveals a significant PL enhancement immediately before the thin film became completely dry (Fig. 2a). Initially and at the conclusion of the experiment, the dry film exhibits only minimal baseline luminescence. Upon solvent addition (at 6 s), the luminescence intensity increases, reaching a temporary plateau (at ~50 s) while the thin film remains fully immersed. During the final stages of drying, as the solvent evaporates from the MTF, the luminescence intensity increases dramatically, reaching a level approximately 50-fold higher than that of the dry state and 10-fold higher than the immersion state. A PLQY of 42.5% was determined at this stage, confirming the absolute increase in PL. The temporal evolution of luminescence, spanning from the initial enhancement to peak intensity and the subsequent decay to the dry state, lasts for approximately 1 min.

To distinguish different phases during solvent evaporation, the complete temporal evolution of the PL spectra is presented in Fig. 2b and Supplementary Fig. 18. Within the 380 to 480 nm spectral region, three dominant peaks are observed. These peaks closely resemble the emission profile of ADC linkers in solution (Supplementary Fig. 19) and their positions shift over time. An initial blue shift occurs immediately following solvent application, with this shift being most prominent for the two highest-energy peaks. This behavior is analogous to minor variations in emission peak positions reported in the literature for other Zn-ADC paddlewheel-based structures in solution and dry states[46,47]. Consequently, this initial peak shift is attributed to the interaction of the linker with the solvent as it gradually fills the MOF pores, and this phase is termed the "immersion" state. Following the onset of immersion, the magnitude of the blue shift remains relatively

constant until another sudden shift occurs during the final evaporation of ethanol at the stage of transient drying, coinciding with the spike in luminescence intensity. This emission phase is designated the "light-up" state, where the abrupt blue-shift suggests an additional effect beyond simple solvatochromism occurring within the MOF during this transient drying stage. At the end of this phase, the system reverts to its initial dry state, as evidenced by the decreasing emission intensity and the return of peak positions to their original values. In addition to the emission between 390 and 450 nm, a very broad emission centered at ~600 nm is observed. This broad emission is slightly stronger than the short-wavelength emission in the dry state (Fig. 2c). Literature reports suggest the existence of various low-energy emissive states in densely packed anthracene-based molecular systems, arising from orientation-dependent energetic coupling of neighboring chromophores[48–50]. We hypothesize that such a state may also form in the Zn-ADC-DABCO system through the reconfiguration of adjacent linkers upon UV excitation[51], leading to a slipped face-to-face orientation of their π planes. This red emission is suppressed during the light-up state and recovers in the dry state, providing an initial indication of a collective realignment of ADC rotors, a phenomenon that will be discussed further. For comparison, the evolution of PL and PLQY was studied for Zn-ADC-DABCO powder samples deposited on substrates. As presented in Supplementary Fig. 20 and Table 1, the oriented MTF exhibits a superior luminescence enhancement in the light-up state compared to the powder counterpart. This finding highlights the role of crystallite orientation in governing the responsiveness to ethanol evaporation, suggesting the behavior of a more vertical tilt of the anthracene rotators at the light-up state.

Regarding fatigue resistance, the luminescence modulation induced by repeated solvent application and evaporation is highly reproducible (Fig. 2d). Furthermore, the structural properties of the MTF, including its orientation, remain constant over these cycles (Supplementary Fig. 21). An increase in luminescence intensity during the light-up state is observed with larger applied droplet volumes (Fig. 2e), suggesting a correlation between the emission turn-on effect and the duration before transient drying stage. Moreover, solvent molecular size plays a critical role in the emission enhancement observed during both the immersion and light-up states. Analogous to ethanol, the smaller methanol molecule induces emission turn-on, whereas the larger isopropanol molecule does not (Fig. 2f). This observation supports the hypothesis that the critical event transpires within the MOF channels and is related to the interaction of solvent molecules with the anthracene moieties of the linkers. Additionally, a similar luminescence turn-on process was also observed in the larger-pillared Zn-ADC-BPy MOF system (Supplementary Fig. 22). Therefore, evaporation of solvent at the transient drying stage reproducibly triggers a significant and reversible modulation of their PL, influenced by solvent volume and molecular size.

## Drying kinetics on MOF thin films

To elucidate the mechanism for PL enhancement at the light-up state, the moieties responsible for optical properties were first confirmed. The absorption spectra of the MTF strongly resemble those of the pure ADC linker in solution (Supplementary Fig. 23), indicating that ADC is the primary component determining the optical properties. Given the structural characteristics of the ADC linker, a conformational deformation of the anthracene rotator during the light-up state is proposed as the reason responsible for suppressing self-quenching. To link this PL enhancement to the drying kinetics, an in-situ experimental setup combining optical measurements and QCM with dissipation monitoring was employed (Supplementary Note 2). Initial vapor sorption experiments were performed to establish a comparison (Supplementary Fig. 24). Upon exposure to vapor, a decrease in frequency ($\Delta f/n$) is observed, indicating mass uptake. Concurrently, the bandwidth increased, indicating a rise in viscous dissipation. A stationary state was

achieved within a few minutes. The values of $\Delta f/n$ exhibit slight variations across different overtones in both the dry and vapor-saturated states, reflecting the finite softness of the sample. The mass of the dry and vapor-exposed states was derived using $-\Delta f/n$ values averaged over the different overtones. This analysis revealed a massive increase of 15±2% upon vapor exposure. This value is lower than the theoretical pore volume of 23% reported previously[52]. This incomplete filling is likely attributable to the polarity mismatch between the polar ethanol guest molecules and the non-polar pores of the MOF. Furthermore, while a minor increase in PL is observed upon the termination of vapor exposure, the magnitude of this effect is significantly lower than the enhancement observed during liquid immersion experiments.

Figure 3 and Supplementary Fig. 25 illustrate the ethanol evaporation kinetics during the liquid immersion experiments. These data confirm that the immersion state itself does not yield an increase in PL. A significant PL enhancement is induced during the transient drying phase. The drying process spans ~1 min and is characterized in the QCM data by an increase in frequency that is approximately linear with respect to time. Notably, both the frequency and bandwidth signals exhibit a temporal drift when the layer is immersed (indicated by the gray arrows in Fig. 3). These sloping plateaus reflect time-dependent changes in the properties of the MTFs during the immersion state, while a behavior is absent in the vapor sorption experiments (Supplementary Fig. 24). This contrast implies that liquid ethanol induces a structural distortion within the MOF layer that vapor exposure does not. Importantly, this distortion is non-destructive and fully reversible. The reverse transformation occurs as the sample dries. It is hypothesized that this reverse transition generates transient internal stress, which drives small-scale rearrangements within the MTFs. These rearrangements effectively suppress self-quenching, thereby triggering the amplification in PL intensity observed during the transient drying stage.

To further validate this mechanism, varying volumes of liquid ethanol were applied to the film. As illustrated in Supplementary Fig. 26, the magnitude of PL enhancement increases with the volume of ethanol, a trend consistent with the results shown in Fig. 2e. The QCM result reveals that larger droplet volumes prolong the immersion duration before the MTF dries. Extended exposure to the liquid ethanol correlates with a more pronounced drift in the QCM frequency and bandwidth signals, indicating more extensive structural perturbations within the film. Consequently, the stress generated during the transient drying stage increases or increasingly spreads across the whole MTF, resulting in a higher degree of PL enhancement. The transient drying process of the liquid experiment requires ~1 min, a kinetic profile that is slower than the desorption observed in vapor-sorption experiments. This suggests that the structural deformation of the MTFs itself acts to retard the drying kinetics. Collectively, these results demonstrate that the MTFs do not function as a static rigid scaffold; its structural properties are dynamic and responsive to ethanol molecule movement inside the structure. During the transient drying stage, the evaporating solvent exerts internal stress on the framework. This stress subsequently drives a structural realignment at the molecular level, likely involving the partial rotation of the central anthracene moiety of the ADC linker.

## Evaporation induces nano rotor alignment at the transient drying stage

To prove that the self-quenching is suppressed at the transient drying stage, time-resolved PL characterization was performed. PL decay characteristics reveal bi-exponential behavior in the dry and immersion states (Fig. 4a and Supplementary Table 2), indicating multiple deactivation pathways for excited chromophores. However, excited-state deactivation appears simplified in the light-up state. The mono-exponential decay characteristic indicates only a single dominant radiative decay channel. Similarly, ADC in ethanol also exhibits a

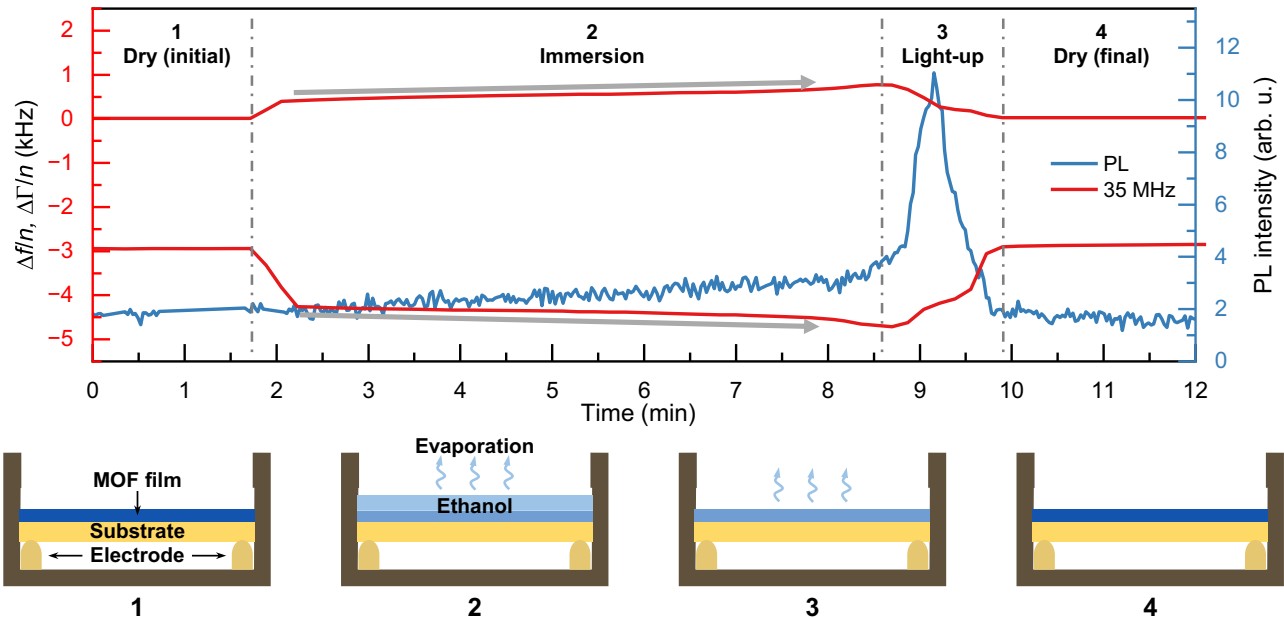

**Fig. 3 | Evolution of QCM and PL signals during ethanol evaporation.** Top part: Shifts in overtone-normalized resonance frequency, Δf/n (negative), and half-bandwidth, ΔΓ/n (positive), upon dropping 20 μL ethanol on the quartz plate. The gray arrows point out the sloped plateaus in the QCM signal, indicating disturbance on the MOF structure. Bottom part: The schematic diagram shows the different states during ethanol evaporation. Source data are provided as a Source Data file.

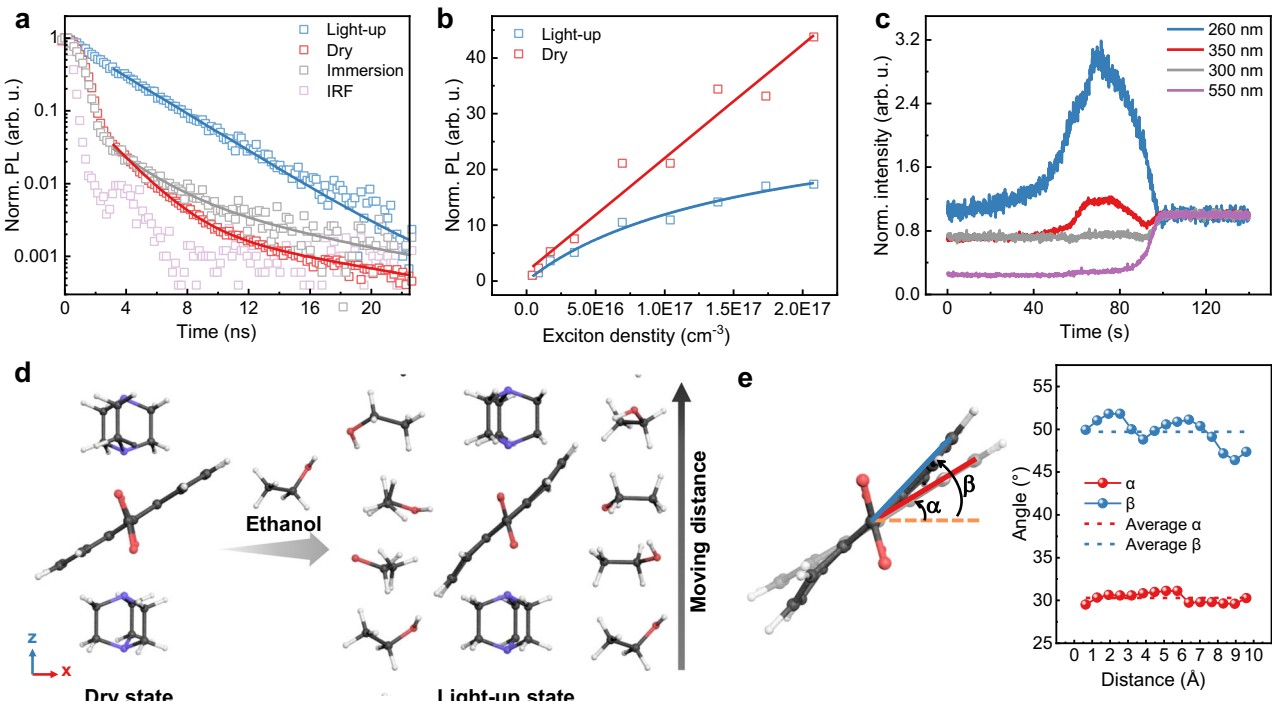

**Fig. 4 | ADC rotation inside the Zn-ADC-DABCO MOF. a** Time-resolved PL of ADC in Zn-ADC-DABCO. The PL signal of different emission states is presented by different colored scatter plots. In an overlay, mono-/bi-exponential tail fits (lines) of the PL decays are shown. **b** PL intensity over excitation fluence in dry state and light-up state. The sublinear increase in the light-up state indicates improved exciton transport that leads to pronounced exciton-exciton annihilation at higher exciton densities. The data shown are the mean values from three solvent evaporation experiments at each fluence. **c** Reflection of Zn-ADC-DABCO thin film on Si during ethanol evaporation at different wavelengths. There is a strong increase before the thin film turns dry again at wavelengths coinciding with the longitudinal and transverse transition dipole moments of anthracene (260 and 350 nm). The individual profiles have been normalized to their final dry state level, and the time axes of each profile have been aligned based on their final dry state transition. **d** The process of ethanol movement from both sides of ADC. Distance is the length of one ethanol molecule moving from bottom to top. Atoms: C, black; N, blue; O, red; H, white. **e** Evolution of the angles of ADC surrounded by moving ethanol (β) and the dry state ADC without ethanol passing through (α). Source data are provided as a Source Data file.

mono-exponential decay profile (Supplementary Fig. 27). This suggests that certain non-radiative pathways, present in the dry and immersion states, become suppressed in the light-up state. Although the inter-linker distances in the light-up state are sufficient to preclude excimer formation, resonant energy transfer can still occur between neighboring anthracene units. As shown in Fig. 4b, whereas the fluence-dependent luminescence in the dry state exhibits a linear characteristic, a sublinear profile is observed in the light-up state. This sublinearity indicates improved exciton diffusion, and simulations based on a one-dimensional anisotropic exciton transport model yield an estimated diffusion length of ~120 nm (Supplementary Note 3)[53]. This enhancement is attributed to the collective alignment of chromophores, which decreases energetic disorder.

The optical and QCM results demonstrate that Zn-ADC-DABCO containing chromophores arranged as paddle wheels exhibit a peculiar type of internal flexibility when exposed to liquid ethanol. To prove the scenario of ADC rotation, thin-film reflection was monitored using orthogonal illumination of the Si substrate. The steady-state reflectance spectrum of the Zn-ADC-DABCO thin film exhibits a pronounced dip in the 350–400 nm range (Supplementary Fig. 28), mirroring the features observed in its absorption spectrum. In the experimental configuration for kinetics measurement, the transition dipole moment of ADC is nearly perpendicular to the incident illumination owing to the <001>-orientation. Therefore, a collective realignment of the linkers would reduce the chromophore absorption cross-section, thereby causing a detectable macroscopic decrease in absorption. When recording the reflection from the MTF on a Si substrate, the detected signal predominantly comprises light that has traversed the MOF twice, with a minor contribution from light scattered at the film surface. Consequently, an increase in the reflection signal signifies a reduction in thin-film absorption. The experimentally obtained temporal reflection signal is depicted in Fig. 4c for different probe wavelengths. For probe wavelengths of 260 and 350 nm (corresponding to the longitudinal and transverse transition dipole moments of anthracene, respectively)[51], a clear increase in the reflection signal is observed immediately before the thin film becomes dry. This increase occurs on a timescale very similar to that of the strong emission enhancement in the light-up state. Conversely, at wavelengths of 330 and 550 nm, where linker absorption is significantly lower, the reflection modulation diminishes to a sudden transition from higher to lower values, primarily reflecting the optical characteristics of the bare Si substrate (Supplementary Fig. 29). In addition, the evolution of absorption was studied for both the oriented Zn-ADC-DABCO MTF and its powder deposited on a quartz plate. As depicted in Supplementary Fig. 30, the oriented MTF exhibits a more significant decrease in absorbance than its powder counterpart. These findings suggest the partial rotation of ADC linkers towards a perpendicular alignment within the oriented MTF, which is in agreement with the conclusions previously derived from the PL behavior.

Molecular dynamics simulations offer deeper insight into the possibility of molecular motion (Supplementary Note 4). Initially, the climbing image nudged elastic band (CINEB) method was employed to simulate the passage of a single ethanol molecule through the MOF channel along the c-axis (z-direction)[54]. As illustrated in Supplementary Fig. 31a and Supplementary Data 1, an ethanol molecule traverses five distinct stages when passing from the entrance to the exit of a MOF channel segment containing four ADC molecules. Upon entry of the ethanol molecule into this segment, two ADC molecules rotate and bend downwards to facilitate its ingress. Conversely, as the ethanol molecule exits, the remaining two ADC molecules rotate and bend upwards. The potential energy profile was calculated for these five stages (Supplementary Fig. 31b). The profile reveals two energy barriers corresponding to the ingress and egress of the ethanol molecule, respectively. This indicates that ADC linkers undergo rotation and bending during ethanol passage and subsequently recover their initial

state after ethanol has exited. Indeed, the continuous evaporation of ethanol can maintain the ADC linkers in a persistently inclined state (Supplementary Data 2). Considering a single ADC molecule within the unit cell, ethanol molecules can evaporate from both sides. Simulations of this scenario show that ADC molecules exhibit pronounced rotation as ethanol passes from both sides (Fig. 4d and Supplementary Data 3). Estimation of the ADC rotational angle relative to the horizontal direction (Fig. 4e) reveals an average angle of 49.7° during ethanol evaporation, compared to an average angle of 30.3° in the dry state. Based on these simulations, it is inferred that the stable deflection of the linker provides a plausible explanation for the observed luminescence turn-on effect. Continuous passage of solvent molecules through the channel subjects the ADC rotors to a persistent torque. Under this steady dynamic regime, the luminescence of the linker can be significantly enhanced[55,56]. It should be noted that the simulation shows a partial rotation[27], which aligns the anthracene unit more vertically, rather than a complete unhindered turnstile rotation. This simulation agrees with the experimental PL enhancement, because complete non-unidirectional rotation characteristically promotes non-radiative decay pathways, thereby resulting in luminescence quenching, as exemplified in the Introduction by the molecules possessing phenyl rotators[20,22]. To verify this hypothesis, the PL evolution of Zn-NDC-DABCO was investigated during ethanol evaporation. The 1,4-naphthalenedicarboxylic acid (NDC) linker is less bulky than ADC, affording smaller steric hindrance and enough void space within the MOFs. As shown in Supplementary Fig. 32, at the transient drying stage, its luminescence is quenched. This quenching is potentially due to the less restricted rotation of the NDC linker.

## Proof-of-concept evaporation indicator

The solvent evaporation sensing capability of the MTF was estimated in the application combining the QCM with PL measurement. Unfortunately, attempts to perform viscoelastic analysis to quantify ethanol evaporation were unsuccessful due to the non-planar geometry of MTFs. Based on the finding that the PL of the MOF reports the internal stress generated during the transient drying stage, a prototypical application of Zn-ADC-DABCO as a volatile solvent evaporation indicator was demonstrated. For this proof-of-concept, the MOF was coated onto 4 Å molecular sieve beads using the drop-casting method (see Methods). When the MOF-coated sieve is positioned on a methanol droplet, liquid absorption into the sieve is initiated. Consequently, the area saturated with methanol becomes a wet region, while the area yet to absorb the liquid remains dry. Continuous solvent evaporation occurs at the interface between these two regions. It is hypothesized that the localized evaporation at this moving boundary generates internal stress, which subsequently activates the luminescence of the MOF. Indeed, the MOF-coated spherical sieve provides an eye-visible optical indication of methanol evaporation (Fig. 5 and Supplementary Movie 1). In this experiment, methanol sorption and evaporation are recorded under 254 nm UV irradiation. To distinguish the position of luminescence and sorption, the upper row of Fig. 5 displays the sieve with room lighting turned off, while the lower row shows the sieve under ambient lighting. Upon initial exposure to methanol (0 s), a distinct luminescent ring emerges on the surface of the sphere. This ring migrates upward, eventually converging into a bright spot at the top. This apical emission persists for several minutes even after the exterior of the sieve appears dry (bottom Fig. 5f). The disappearance of PL suggests the cessation of solvent evaporation from the interior of the sieve. This proof-of-concept demonstrates the potential of Zn-ADC-DABCO as a functional coating for the real-time visual indication of solvent evaporation.

## Discussion

In summary, a nanoscale luminescent indicator for evaporation flow has been developed, functioning through the dynamic realignment of

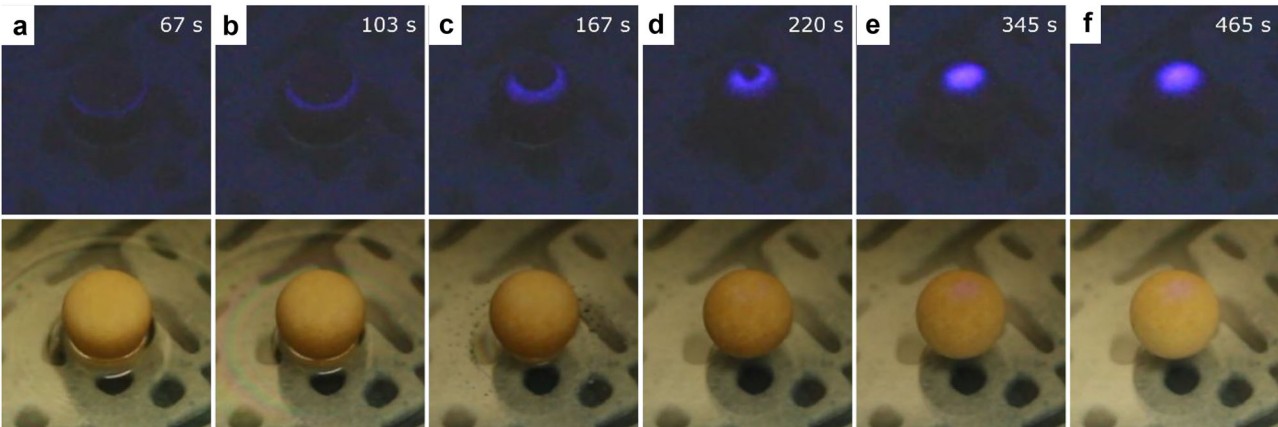

**Fig. 5 | Zn-ADC-DABCO thin film on molecular sieve indicates methanol evaporation. a–f** Photo series of a MOF-coated molecular sieve under 254 nm excitation with room lighting off (top row) and room lighting on (bottom row). The time stamps indicate the elapsed time after exposure to the solvent. The distinctly shaped luminescence on the surface of the sieve indicates moving channels of preferred solvent evaporation from the sieve. The confined emission at the top remains for several minutes, reporting continued solvent evaporation even after the sieve appears dry already on the outside.

linkers within an oriented MTF. These films are readily fabricated by drop-casting a precursor solution onto various substrates (including Si, $Gd_2O_3$@Si, $TiO_2$@Si, ZnO@Si, Ag@Si, and quartz) maintained at 50 °C. The observed order-of-magnitude contrast in PL intensity between the dry and transient drying states confirms that the final stage of solvent evaporation triggers a realignment of chromophores within the MTFs. The ability of molecular rotors to signal conformational transitions establishes a foundation for the development of nanoscale evaporation indicators. Moreover, the underlying sensing mechanism based on molecular conformational changes has implications extending well beyond solvent evaporation. For instance, guest transport within the MTF pores may induce a change of internal molecular strain and rotor rearrangement, paving the way for the development of nanoscopic flow sensors. Future investigations should aim to expand the library of compatible analytes and establish quantitative stimuli-response dependencies via structural optimization (e.g., tuning pore size and ligand chemistry). This work is expected to stimulate further research into the real-time tracking of volatilization and fluid transport through molecular-scale pores.

## Methods
### Materials
All reagents and solvents were used as received or otherwise indicated. The chemical precursors used were purchased from Alfa Aesar (Zn acetate dihydrate (>97 %), BPy (98%)), Sigma-Aldrich (DABCO ( ≥ 99 %), ethanolamine (>99.5%), n-hexane (≥95 %), acetone (for analysis)), BLD-Pharm (ADC (98 %), and Merck (methanol (EMPLURA), ethanol (EMPLURA), cyclohexane (≥99 %), toluene (≥99.7 %)). $Gd_2O_3$@Si substrates with 150 nm $Gd_2O_3$ coated by molecular-beam epitaxy were purchased from Translucent Energy, Inc. Gold-coated quartz crustal substrates (BT-QC05QSG, 5 MHz, 14 mm) were purchased from BeamTec GmbH.

### Synthesis of MTF on various substrates
The substrates used for drop-casting were rinsed with ethanol and dried with compressed air before pre-heating on the hot plate to 50 °C. The typical substrate size was 0.5 cm×1 cm. In order to achieve <001 > -oriented DABCO-based MTF, the metal precursor, ADC and DABCO, respectively, are mixed in ethanol at 0.15 mM concentration. The best films based on BPy as a pillar molecule were achieved with a metal and ADC precursors at 0.2 mM with BPy at 0.05 mM. The metal precursor solution is produced by dissolving Zn acetate dihydrate in ethanol with HCl(aq) in an equimolar ratio, and kept as a stock solution. HCl(aq) stabilizes the Zn acetate phase and prevents the formation of Zn

hydroxide compounds. Explicitly, this means for the synthetic protocol that there is always an equal amount of Zn acetate and HCl(aq) molecules present in the precursor solution. The Zn solution is added to the growth solution for synthesis immediately before the drop-casting to avoid pre-crystallization in the solution. The droplet size should be chosen such that it can still be supported by surface tension on the substrate. After every drop-casting, the samples are washed by shaking in an ethanol bath, followed by compressed air drying.

### Synthesis of MOF material on molecular sieves
The 4 Å molecular sieves ($Na_{12}[(AlO_2)_{12}(SiO_2)_{12}]\cdot xH_2O$, Merck) were coated using the same method. For better heat conduction between the hot plate and the spherical sieves, they were partially wrapped with aluminum foil. Ethanol-based precursor solution droplets were then applied at the top of the sieves, flowing around them in the process. After multiple dropping cycles, the sieves were taken out of the foils and washed in an ethanol bath, followed by a methanol bath and then left for drying.

### Synthesis of ZnO film on Si substrate
28.1 mg ethanolamine and 101 mg zinc acetate dihydrate were dissolved in 1 mL 2-methoxyethanol upon stirring at 70 °C for 24 h. The precursor solution was then spin-coated at 4000 rpm for 30 s (ramp rate 1000 rpm s$^{-1}$), followed by the post annealing at 300 °C for 30 min.

### Deposition of $TiO_2$ layer on Si substrate
Atomic layer deposition (ALD)-$TiO_2$-coated silicon substrates were prepared by using a Picosun R-series ALD reactor. The deposition was performed following parameters: temperature 300 °C, number of cycles 125, pulsing and purging time for $TiCl_4$ 0.1 s and 9 s, respectively; while for water 0.1 s and 13 s, respectively. The thickness of $TiO_2$ on a Si substrate is ~3 nm.

### Deposition of Ag layer on Si substrate
The magnetron sputtering of the Ag layer was carried out at the Emitech K575XD (Quorum Technologies Ltd, UK). The sputtering was made at 90 mA for 100 s with 1 cycle. The examined thickness of the Ag film is ~22 nm.

### Structural and composition characterization
The PXRD data were taken on a Bruker D2 Phaser (primary track: unpolarized Cu K-alpha X-ray source (30 kV, 10 mA), equatorial Soller slit, horizontal 1 mm slit; secondary track: equatorial Soller slit,

LYNXEYE strip detector, sample-to-detector distance: 141.0 mm) with the thin film samples mounted on a rotating sample stage. The GIWAXS data were acquired with a Bruker D8 Advance (primary track: unpolarized Cu K-alpha X-ray source (40 kV, 40 mA), Goebel mirror, 0.5 mm micromask, 0.3 mm snout; secondary track: DECTRIS Eiger2 R 500 2D detector, sample-to-detector distance: 118.1 mm). Grazing-incidence angles from 0.2° to 0.6° and exposure times from 1800 s to 43200 s were chosen based on the intensity of the diffraction signal created by each individual sample.

The SEM images were acquired with a ZEISS Supra 60VP with the operating parameters for top view: working distance 7.9 mm, electron high tension 10 kV, magnification 50000; and for cross-sectional view: working distance 10.7 mm, electron high tension 3 kV, magnification 50000. In order to record the cross-section, the sample was broken in half at the center. Prior to taking the SEM images, the sample was coated with a 5 nm silver layer.

FTIR-ATR spectra were carried out on a VERTEX 70 FT-IR spectrometer (Bruker). X-ray photoelectron spectroscopy (XPS) was carried out with the dynamic scanning X-ray photoelectron microprobe (PHI Quantes), having a pass energy of 280 eV, energy steps of 1 eV each increment, and a dwell duration of 100 ms/step for each energy step for Survey scans. The peak fits were done on high-resolution XPS spectra with a pass energy of 26 eV for the host material and 55 eV for the dopants. The energy per step was 0.1 eV, and the time per step was 20 ms.

### Photophysical characterization
Steady-state PL spectra, time-resolved thin film reflection during ethanol evaporation and ultrafast time-resolved PL via TCSPC were taken with an Edinburgh Instruments Fluorescence Spectrometer FS5 equipped with dedicated sample holders to measure thin films and liquids. To capture the steady-state emission, a Xe lamp and monochromator were used as an excitation source with a suitable long-pass filter on the secondary side to eliminate excitation light in the emission path. For the TCSPC measurements, an external picosecond light-emitting diode operating at 365 nm was used for excitation, and the emission was captured with a center wavelength of 430 nm. To be able to record TCSPC data of the sample in the light-up phase, the very short time duration of the strong PL turn-on needed to be artificially extended. To achieve this, after applying an ethanol droplet, the MTF was covered with a quartz substrate and sealed with a PTFE foil around the edges right at the onset of the strong PL enhancement. With this approach, the sample could be kept in the bright turn-on phase for several minutes. To also extend the duration of the immersion phase for the TCSPC measurement by several minutes, the sample was put into a flat-lying, open 1 cm-by-1 cm quartz cuvette before applying the ethanol on its surface. Time-resolved thin film reflection during the evaporation of ethanol was performed with the sample lying flat on a horizontal thin film holder, such that illumination and collection occur from vertically above the substrate. Excitation and emission wavelengths were set to the same values while protecting the detector from too high signal intensity with an appropriate long-pass filter on the emission side. The measurement was started after the solvent droplet was applied to the sample.

The steady-state UV-Vis absorbance spectra of ADC in solution and MTF on quartz substrate, as well as reflectance spectra of MTF on Si substrate, were obtained using an Agilent Cary 7000 Spectrophotometer with a dual light source consisting of a quartz tungsten halogen visible and deuterium arc for the UV. The instrument was equipped with an integrated sphere with a highly reflective inner surface (Spectralon) to collect both diffused and direct light.

The time-resolved fluence-dependent PL measurements during solvent evaporation from Zn-ADC-DABCO were conducted with a self-assembled setup utilizing a fs-pulsed laser with an external harmonic generator (LightConversion Pharos and HIRO) to create an excitation source at 343 nm and 20 kHz repetition rate. The emission from the sample was recorded with a fiber-coupled (Thorlabs FP1000URT) Ocean Insight QEPro spectrometer. A second experimental setup for additional solvent evaporation PL measurements and PLQY comprised collimated UV LED excitation (Thorlabs M300L4 and M365LP1) and a fiber-coupled (Thorlabs BFY400MS02) spectrometer (AvaSpec-ULS2048x64TEC-EVO, Avantes). For emission detection, two lenses were used to collimate and focus the light onto the fiber with long-pass filters to reject the UV excitation. The emission spectra were acquired over time by means of a home-made automated LabVIEW control software. For the PLQY measurements, the samples were mounted in a 15 cm integrating sphere, and the data were acquired based on the 3 M method, presented by de Mello et al. [57]. The temperature-dependent PL lifetime was acquired using a streak camera (Hamamatsu Universal C10910) equipped with a spectrometer (Acton SpectraPro SP2300). The temperature of the sample was controlled in an Oxford NanoScience OptistatDry TLEX Cryostat with liquid helium. The measurements were conducted in a vacuum.

### Reporting summary
Further information on research design is available in the Nature Portfolio Reporting Summary linked to this article.

## Data availability
Source data are provided with this paper. The data generated in this study are provided in the Supplementary Information/Source Data file. Any additional information can be obtained from the corresponding authors. Source data are provided with this paper.

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

## Acknowledgements

The authors express their gratitude to the DFG for funding through the priority program SPP 1928 COORNETS, as well as the Helmholtz Association for funding through (i) HEMF, (ii) the MTET program (Materials and Technologies for the Energy Transition)—Topic 1—Photovoltaics (38.01.02), and (iii) the recruitment initiative of B.S.R. We, furthermore, acknowledge the support of the Karlsruhe School of Optics and Photonics (KSOP) and the Ministry of Science, Research and the Arts of Baden-Württemberg (Excellence Initiative II). We also appreciate the support of the National Natural Science Foundation of China (12274097, L.G.). We thank Justine Nyarige (KIT) for supplying $TiO_2$@Si substrates. We thank Dr. Arne Langhoff (TU Clausthal) for helpful discussions.

## Author contributions

T.H.Z. and I.H. conceived the idea and supervised the project. J.F., T.H.Z., I.A., B.S., and D.J. designed the experiments. J.F. and T.H.Z. designed, synthesized, and characterized the MOF thin films. Y.L. prepared the glass substrates. J.F., T.H.Z., N.R., and D.B. performed the photophysical studies. P.S., and T.H.Z. carried out the QCM measurements. H.J., T.Z., L.G., and P.D. performed the molecular dynamics simulation. E.H. carried out the XPS measurements and participated in data analysis. J.F., T.H.Z., and D.J. wrote the manuscript. All authors analyzed and discussed the results and have given approval for the final version of the manuscript.

## Funding

## Competing interests

The authors declare no competing interests.
