## [Transparent Peer Review file · Nature Communications]

Turn-on Luminescence from Molecular Rotor Realignment in Metal-Organic Framework Thin Films

Corresponding Author: Dr Tonghan Zhao

Version 0:

Reviewer comments:

Reviewer #1

(Remarks to the Author)

In this manuscript, a highly oriented columnar layer MOF film (Zn-ADC-DABCO film) was synthesized using a simple and fast drop casting method, and the film formation was demonstrated by SEM, GIWAXS, FWHM, etc. The experimental results show that during the solvent evaporation process, the PL is significantly enhanced immediately before the film is completely dried. This is a nanoscale luminescent indicator for evaporation flow, and its working principle is achieved through the dynamic rearrangement of the connectors in the oriented MOF film. Despite a lot of proof, there are still some problems in this manuscript that require additional explanations and tests from the authors. Overall, I believe that once these issues are clarified, the work will be suitable for publication. The details are as follows:

1. In the introduction of the manuscript, the authors mentioned that the MOF combined with molecular rotors will be loaded into the film. Therefore, it is recommended that the authors add relevant content in the introduction on "the advantages of metal organic frameworks (MOFs) and MOF films as functional carriers" to introduce the unique properties of MOFs materials, emphasize the advantages of MOF films and review the research status of the combination of molecular rotors and MOFs.
2. In the introduction of the manuscript, it is necessary for the authors to explain in detail the structural characteristics, advantages and working principles of molecular rotors, and review the research progress and current status of molecular rotors in flow sensing and other aspects.
3. In addition, the introduction still needs to briefly explain the key role of nanoscale flow in microfluidic chips, biomedicine, materials science and other fields, point out the limitations of traditional flow detection methods at the nanoscale, and emphasize the urgent need to develop new nanoscale flow indication tools.
4. At the end of the introduction, it is recommended that the authors add a scheme diagram to briefly summarize the synthesis method, highlights and mechanism of this manuscript.
5. In the section "Oriented metal-organic framework thin films fabricated by drop-casting" in the manuscript, I think it is necessary for the authors to supplement the X-ray photoelectron spectroscopy (XPS) and infrared spectroscopy (IR) before and after testing the loading to analyze the elemental composition and chemical valence state of the film surface, and to determine whether there is an interaction between MOF and the film based on the characteristic absorption peaks and absorption bands.
6. As a potential nanoscale flow indicator, it is necessary to have good stability. To this end, it is recommended that the authors supplement the PXRD pattern of Zn-ADC-DABCO film placed in air and ethanol solution for 5 days.
7. In the "Luminescence turn-on effect" section of the manuscript, the authors mentioned that the "flow" state caused the luminescence to turn on and off. Could this change be due to the ethanol molecules entering the MOF pores, thereby restricting the rotation of the molecular rotor, thereby causing Zn-ADC-DABCO to exhibit bright emission? Rather than the flow state? What is the difference between the "flow state" mentioned by the authors and the AIE mechanism?
8. In addition to the ADC linkers mentioned in this manuscript, can the authors provide films formed by similar linkers to prove the author's claim that "dynamic rearrangement of the linker achieves luminescence difference"? I think this manuscript should also provide films with molecular rotors, films without molecular rotors, and films formed by other molecular rotors for simultaneous control tests. It is far from enough to explore only one example.
9. In fact, this "flow state" is related to the pores of MOF. It is recommended that the authors conduct additional testing on the pore characteristics of the Zn-ADC-DABCO film.
10. There are some minor errors in the reference format. Please check the formatting again.

Reviewer #2

(Remarks to the Author)

This manuscript reports a solvent evaporation-induced alignment of ADC rotors within a MOF thin film, resulting in enhanced photoluminescence. The authors utilize a hot-plate drop-casting method to fabricate <001>-oriented Zn-ADC-DABCO MOF films, and employ GIWAXS, reflection spectroscopy, PL measurements, and molecular dynamics simulations to investigate the solvent-driven rotor alignment mechanism. They further propose potential applications in nanoscale fluid sensing. While the work offers practical value in thin-film fabrication, it suffers from significant limitations in scientific novelty, mechanistic depth, and application feasibility:

- The oriented Zn-ADC-DABCO MOF film and its PL property have been well reported in other paper, which lead to a lacking novelty in this work.
- The reported "flow sensing" phenomenon is based solely on solvent evaporation, representing a photophysical change rather than true dynamic fluid detection.
- The mechanism reflects an incidental conformational change triggered by hydrophobic solvent molecules, lacking a generalizable design principle or broader applicability.
- Similar phenomena involving rotor-induced luminescence modulation in MOFs under pressure or solvent stimuli have been previously reported; this work does not constitute a substantial advancement in mechanism or functionality.
- The claimed device application is overstated since the "flow state" response is transient (lasting only a few seconds), with no strategy to achieve stable or tunable signal output.

Even beyond these major concerns, there are additional technical issues that would need to be addressed should the authors consider resubmission elsewhere

In addition, there are some specific suggestions and issues that need to be modified

1. phase angle (azimuthal) measurements are recommended to provide more direct and comprehensive evidence of the film orientation. This would better highlight the degree of crystallographic alignment.
2. The thin-film structure is confirmed only by simulated GIWAXS patterns from literature data. No single-crystal XRD or CCDC deposition is provided, limiting structural certainty. Structural refinement is recommended.
3. The CI-NEB simulation in Supplementary Note 3 yields a very low rotational barrier, but no temperature-dependent PL lifetime or activation energy measurements are provided for validation.
4. The observed PL enhancement during the "flow state" is based only on intensity changes, with no absolute PLQY measurement. Integrating sphere measurements are recommended.
5. The "flow state" lasts only a few seconds, which limits practical applications. Strategies to extend or control the response duration should be proposed.
6. The rotor angle change is derived entirely from simulations (49.7° vs. 30.3°). No direct experimental evidence such as solid-state NMR, EXAFS, or single-crystal XRD is provided.
7. Although GIWAXS after solvent cycling suggests structural retention, there is no BET or gas adsorption data to confirm pore integrity. Additional porosity characterization is recommended.

In conclusion, I do not recommend this manuscript for publication in Nature Communications.

Reviewer #3

(Remarks to the Author)

This manuscript presents an investigation of a MOF used as a photoluminescent (PL) probe during solvent evaporation. The authors describe a simple drop-casting procedure to fabricate oriented films of this MOF, which are then employed to monitor PL intensity changes during evaporation. They attribute the observed increase in PL to a flow-induced alignment of molecular rotors.

The work is overall interesting, particularly in its attempt to correlate evaporative flow with optical changes in a MOF. However, several critical aspects require clarification and evidences for instance on the relevance of this system for real applications. The key claims, especially regarding the sensing mechanism and the MOF orientation process, are not sufficiently supported by experimental data. Below I provide detailed comments and suggestions for improvement:

- 1) One of the main issue is related to the relevance of this approach. Beyond the interesting observation, the practical relevance of the proposed approach is not clearly justified. Beyond ethanol (and possibly methanol), it is unclear how this could translate into real applications. What would be the envisioned device? For which solvents? A more in-depth discussion on potential use cases or limitations would be valuable.
- 2) On the oriented MOF: the ability to obtain oriented Zn-ADC-DABCO films by simple drop casting is a potentially impactful claim. Orientation in MOFs is typically non-trivial and often requires external stimuli or epitaxial growth. Two mechanisms are discussed in SI Figure S6. Mechanism (b) appears more plausible, especially since orientation occurs on various substrates. Is this driven by sedimentation, or by capillary forces during drying? Could pre-crystallization in solution followed by controlled evaporation play a role? In this case this would resemble to other previously reported approaches (Mater. Chem. Front., 2023,7, 5545-5560) Experiments that address these points are necessary to support this claim
- 3) page 3 The authors assess orientation via the FWHM of the {112} diffraction peak. However, changes in synthesis conditions likely affect crystal size, not just orientation. The crystal size should be characterized. A more robust method could involve comparing intensity ratios of multiple peaks.
- 4) The porosity of the MOF film is not quantified, yet it is central to the flow and PL mechanism proposed. What is the pore size distribution, porous volume % and the interparticle void fraction in the film? Since scattering is later claimed to be

negligible, techniques like ellipsometric porosimetry or krypton adsorption on thin films could be employed to provide these informations.

5) Figure 2 No comparison is shown with non-oriented or powder-based MOF films during solvent evaporation. This is essential to confirm either that the PL enhancement arises from orientation, and not simply from solvent removal or to justify the use of oriented MOFs. Similarly, the reflectivity experiments later in the manuscript would benefit from such control samples.

6) Page 5 The observation that larger droplet volumes yield higher PL intensity during the "flow state" is inconsistent with the proposed mechanism. Assuming a fluid flow inside pores, the amount of ethanol entering the pore system should not scale with the droplet volume. At most, larger droplets should extend the duration of the evaporation process, not increase the amplitude of the PL response (the number of molecular rotors is the same). Please clarify or reconsider this interpretation.

7) Page 6 The interpretation of reflectivity changes as purely due to reduced absorption is not fully convincing. Interferometric effects due to changes in refractive index or film thickness (or even thin liquid layers forming on top) may strongly contribute: the reflectivity modification due to interferences is most pronounced at shorter wavelengths. Please provide full reflectance spectra and discuss these alternative explanations. How is the non-flat liquid interface during evaporation handled for reflectivity? also for experiments on Si substrates? Absorption spectroscopy on transparent substrates could help resolve this. And in any case, reference experiment with not oriented MOF should be provided to support the claim.

8) The discussion on internal flows is currently a little simplified. While flow during evaporation can occur, microporous materials like MOFs can also exhibit vapor-phase transport (e.g., Knudsen diffusion) rather than liquid capillary flow, depending on the pore size, connectivity, and degree of saturation. To probe that, I recommend to perform sorption analysis on the films (ellipsometry porosimetry for instance) in presence of EtOH vapors to provide EtOH desorption isotherms and eventually thickness contraction during desorption. For instance, during evaporation huge capillary force due the formation of a meniscus into the pores can also play a role on the modification of molecules alignment.

9) the experiment with the spherical molecular sieves should be better described and justified. It is not clear (to me at least) where the PL MOFs is located and why the luminescence is observed only on the top. Why this system was chosen as proof of concept? I'm still unconvinced about its relevance. As also in my comment 1, I suggest clarifying this point or provide a more relevant example.

Version 1:

Reviewer comments:

Reviewer #1

(Remarks to the Author)

I am satisfied with the author's revisions. Agree to accept and publish.

Reviewer #2

(Remarks to the Author)

I think the authors have addressed the review comments.

Reviewer #3

(Remarks to the Author)

I have carefully analyzed the revised version of the manuscript as well as the authors' responses to the reviewers. I appreciate the authors' efforts to revise the manuscript and to shift the focus toward the development of evaporative sensors rather than flow sensors. However, several important concerns remain insufficiently addressed, and some responses are unsatisfactory and I am still not convinced of its relevance or of some of the main scientific claims.

More specifically:

point 1 on relevance. The added paragraph regarding potential applications remains overly generic and does not convincingly demonstrate the broader relevance of the device beyond ethanol evaporation detection (is it very relevant problem?). As it stands, the justification and versatility of the method appears speculative and lacks concrete examples, which weakens the argument for publication in Nature Communications.

Point 4 on the porosity.

The QCM measurements indicate adsorption and desorption of ethanol; however, they do not provide meaningful insight into pore size distribution or interparticle porosity, that as I indicated in my initial review, both of which may strongly influence the photoluminescence (PL) response during evaporation. Presenting the data in the form of adsorption isotherms may clarify this point and help determine whether porosity affects the sensing mechanism.

point 6 on Mechanism of PL Enhancement. The new experiment does not satisfactorily address my earlier concern. If the

proposed mechanism is correct—namely, that the PL enhancement originates from internal stresses generated during evaporation—then the PL intensity should be independent of droplet volume. The magnitude of internal stress at a given moment should not scale with the total amount of evaporating liquid. The results shown in Figure S26 do not support the proposed mechanism and therefore leave a core claim unvalidated.

point 7 on Reflectivity Measurements. According to the authors, full reflectance spectra were impossible to collect due to the instrument limitation. While I understand that some characterization techniques may be challenging, reflectivity measurements are not among them, as they are routine in many laboratories. Given that the manuscript discusses optical film properties and they presented reflectivity data at different wavelength (e.g., Figure 4c), such measurements are necessary to substantiate the claims. Their absence remains a significant gap.

point 8 similar to point 4.

Version 2:

Reviewer comments:

Reviewer #3

(Remarks to the Author)

I thank the authors for their response and their efforts in improving this work. Considering all their answers and modifications, I think that the manuscript can be published.

RESPONSE TO REVIEWERS' COMMENTS

Reviewer #1 (Remarks to the Author):

In this manuscript, a highly oriented columnar layer MOF film (Zn-ADC-DABCO film) was synthesized using a simple and fast drop casting method, and the film formation was demonstrated by SEM, GIWAXS, FWHM, etc. The experimental results show that during the solvent evaporation process, the PL is significantly enhanced immediately before the film is completely dried. This is a nanoscale luminescent indicator for evaporation flow, and its working principle is achieved through the dynamic rearrangement of the connectors in the oriented MOF film. Despite a lot of proof, there are still some problems in this manuscript that require additional explanations and tests from the authors. Overall, I believe that once these issues are clarified, the work will be suitable for publication. The details are as follows:

We thank the reviewer for the much-valued suggestions, which have enabled us to improve the manuscript. Following are the detailed actions taken in light of the reviewer's comments. The revisions made in the revised manuscript are highlighted.

1. In the introduction of the manuscript, the authors mentioned that the MOF combined with molecular rotors will be loaded into the film. Therefore, it is recommended that the authors add relevant content in the introduction on "the advantages of metal organic frameworks (MOFs) and MOF films as functional carriers" to introduce the unique properties of MOFs materials, emphasize the advantages of MOF films and review the research status of the combination of molecular rotors and MOFs.

Response: We thank you for your suggestion. We have included the advantages of MOFs and MOF thin films (MTFs) in the Introduction. This reads as follows (page 2, line 4):

“However, solid-state molecular motion is considerably challenged by the spatial confinement, intrinsic rigidity, and intermolecular interactions characteristic of crystal lattices.^{8, 9} Metal-organic frameworks (MOFs), constructed from organic linkers and metal ions, provide ideal platforms for the precise spatial arrangement of mobile molecular components. The highly porous cavities within MOFs generate significant internal free volume, which accommodates translational and rotational degrees of freedom. Additionally, the porosity of MOFs permits the diffusion of guest molecules, thereby facilitating access for external chemical species to interact with and modulate the internal molecular dynamics.¹⁰⁻¹³”

References:

“8. Jiang, P., Jin, M. Design of molecular crystals toward crystalline molecular machines: Rotors, gears, and motors. *ACS Nanoscience Au* DOI: 10.1021/acsnanoscienceau.1025c00109 (2025).

9. Ahmed, E., Karothu, D. P., Naumov, P. Crystal adaptronics: Mechanically reconfigurable elastic and superelastic molecular crystals. *Angew. Chem. Int. Edit.* **57**, 8837-8846 (2018).”

Page 2, line 37: “This principle provides an opportunity for the development of reversible optical sensing functionalities. For these applications, the deposition of MOFs as thin films is a particularly promising strategy.²⁸⁻³¹ A critical determinant of performance in these MOF thin films (MTFs) is their preferred crystallite orientation, which modulates host-guest interactions and optical properties.^{32, 33}”

References:

“28. Falcaro, P., *et al.* Centimetre-scale micropore alignment in oriented polycrystalline metal-organic framework films via heteroepitaxial growth. *Nat. Mater.* **16**, 342-348 (2017).

29. Shekhah, O., *et al.* Growth mechanism of metal-organic frameworks: Insights into the nucleation by employing a step-by-step route. *Angew. Chem. Int. Edit.* **48**, 5038-5041 (2009).

30. Zhang, J. B., Tian, Y. B., Gu, Z. G., Zhang, J. Metal-organic framework-based photodetectors. *Nano-Micro Lett.* **16**, 253 (2024).

31. Li, H., *et al.* Two-dimensional metal organic frameworks for photonic applications [invited]. *Opt. Mater. Express* **12**, 1102-1121 (2022).

32. Velásquez-Hernández, M. D., *et al.* Fabrication of 3D oriented MOF micropatterns with anisotropic fluorescent properties. *Adv. Mater.* **35**, 2211478 (2023).

33. Yang, X. X., *et al.* Chiral liquid crystalline metal-organic framework thin films for highly circularly polarized luminescence. *J. Am. Chem. Soc.* **146**, 16213-16221 (2024).”

2. In the introduction of the manuscript, it is necessary for the authors to explain in detail the structural characteristics, advantages and working principles of molecular rotors, and review the research progress and current status of molecular rotors in flow sensing and other aspects.

Response: Thank you for your suggestion. We have added further details about molecular rotors in the Introduction. It reads as follows (page 2, line 15):

“Within these integrated architectures, molecular rotors are typically incorporated as linkers, wherein the stator component coordinates with metal ions to construct the MOF. These materials exhibit internal dynamics and respond to environmental stimuli, including changes in pressure, temperature, viscosity, and guest molecules.¹⁴ This responsiveness highlights the potential of these dynamic systems for applications in gas adsorption, molecular recognition, drug delivery, and water desalination.¹⁵”

Reference:

“14. Feng, L., Astumian, R. D., Stoddart, J. F. Controlling dynamics in extended molecular frameworks. *Nat. Rev. Chem.* **6**, 705-725 (2022).

15. Dong, J. Q., Wee, V., Peh, S. B., Zhao, D. Molecular-rotor-driven advanced porous materials. *Angew. Chem. Int. Edit.* **60**, 16279-16292 (2021).”

3. In addition, the introduction still needs to briefly explain the key role of nanoscale flow in microfluidic chips, biomedicine, materials science and other fields, point out the limitations of traditional flow detection methods at the nanoscale, and emphasize the urgent need to develop new nanoscale flow indication tools.

Response: We thank the reviewer for this constructive suggestion. Following the comments of all reviewers and our re-evaluation of the term "flow state", we have decided to revise it to avoid potential ambiguity. We attribute the luminescence enhancement to internal stress exerted by the liquid ethanol upon the MTF through investigation on quartz crystal microbalance (QCM) with dissipation monitoring, This stress induces rotation of the ADC linkers during the final phase of ethanol evaporation (we also termed the "transient drying" stage), thereby precipitating a dramatic increase in luminescence intensity. We have re-designated this condition as the "light-up" state in the revised text. Therefore, we think that an introduction focused on luminescent MOF rotors and their potential applications is better suited to address this suggestion. The revisions to the Title, Abstract, and Introduction are shown below:

Title:

“Nanoscale **Evaporation** Indicator: Molecular Rotor Realignment Modulates Luminescence in Metal-Organic Framework Thin Films”

Abstract:

“Sub-unit motion within metal-organic frameworks (MOFs) offers unique opportunities for nanoscale sensing. However, achieving controlled rotation of bulky linkers remains a significant challenge. In this study, a 50-fold luminescence enhancement is observed from a MOF thin film when intra-pore solvent flow orients

the linker chromophores. These MOF thin films can be prepared via a facile drop-casting method on various substrates. The MOF structure consists of zinc-coordinated layers containing rotatable chromophores, separated by pillar molecules. Grazing-incidence wide-angle X-ray scattering analysis confirms the formation of highly oriented films. The deposition of a volatile organic compound, such as ethanol, triggers a significant enhancement in luminescence as the solvent nears complete evaporation. Photophysical characterization and quartz crystal microbalance measurements reveal that this phenomenon is driven by internal stress generated during the final stages of evaporation. This stress probably induces a realignment of the MOF chromophores at the molecular scale. Consequently, this dynamic turn-on luminescence behavior establishes a foundation for nanoscale platforms capable of indicating solvent volatilization in real time.”

Page 2, line 22:

“Among these, luminescent molecular rotors represent a promising class of linkers, characterized by their ability to respond to external stimuli through distinct changes in photoluminescence (PL) properties.^{20, 21} Particular attention has been paid to rotor ligands that exhibit aggregation-induced emission (AIE). For instance, Zhao *et al.* reported a series of tetraphenylethylene-based MOFs that demonstrate turn-on fluorescence in response to volatile organic compounds, temperature variations, and viscosity changes.^{22, 23} Recently, Zhu *et al.* synthesized a triphenylamine-based lanthanide MOF wherein the distortion of molecular rotors is regulated by temperature, thereby inducing multicolor luminescence switching.²⁴ In these reported systems, phenyl rings function as the rotators, a mechanism that requires only a relatively small free volume. However, incorporating bulkier units, such as anthracene, significantly increases the rotational barrier. This steric hindrance can lead to restricted rotation or even stabilize the molecule into a stationary conformation.²⁵⁻²⁷”

References:

“20. Zhang, T., *et al.* Pressure-modulated host guest interactions boost effective blue-light emission of MIL-140A nanocrystals. *Nano-Micro Lett.* **18**, DOI: 10.1007/s40820-40025-01917-40828 (2025).

21. Shustova, N. B., *et al.* Phenyl ring dynamics in a tetraphenylethylene-bridged metal-organic framework: Implications for the mechanism of aggregation-induced emission. *J. Am. Chem. Soc.* **134**, 15061-15070 (2012).

22. Dong, J. Q., *et al.* Aggregation-induced emission-responsive metal-organic frameworks. *Chem. Mater.* **32**, 6706-6720 (2020).

23. Zhang, M., *et al.* Two-dimensional metal-organic framework with wide channels and responsive turn-on fluorescence for the chemical sensing of volatile organic compounds. *J. Am. Chem. Soc.* **136**, 7241-7244 (2014).

24. Wang, H. L., *et al.* Smart lanthanide metal-organic frameworks with multicolor luminescence switching induced by the dynamic adaptive antenna effect of molecular rotors. *Adv. Mater.* **37**, DOI: 10.1002/adma.202502742 (2025).

25. Horike, S., *et al.* Dynamic motion of building blocks in porous coordination polymers. *Angew. Chem. Int. Edit.* **45**, 7226-7230 (2006).

26. Kuc, A., Enyashin, A., Seifert, G. Metal-organic frameworks: Structural, energetic, electronic, and mechanical properties. *J. Phys. Chem. B* **111**, 8179-8186 (2007).

27. Gonzalez-Nelson, A., Coudert, F. X., van der Veen, M. A. Rotational dynamics of linkers in metal-organic frameworks. *Nanomaterials* **9**, 330 (2019).”

4. At the end of the introduction, it is recommended that the authors add a scheme diagram to briefly summarize the synthesis method, highlights and mechanism of this manuscript.

Response: We thank the reviewer’s suggestion. We have revised Fig. 1 to summarize the synthesis method, highlights, and mechanism. The new version is as follows:

Fig. 1 Drop-casting of Zn-ADC-DABCO layer-pillar MOF thin films on hot substrate. **a**, Ethanol growth solution, containing Zn precursor, ADC linker and DABCO pillar, drop-cast on a heated substrate results in MTF growth with preferentially $\langle 001 \rangle$ -oriented MOF structure. **b**, Ethanol evaporation induces significant PL enhancement of MTF. The Zn-ADC-DABCO MTF is shown in the immersion state and light-up state. In the initial dry state, the MOF pores are empty and ADC linkers in their relaxed configuration as show in **a**. When ethanol is added, the solvent molecules diffuse into the MOF cavities. In the light-up state, the final evaporation of ethanol generates local stress, deforming the MTF on a molecular scale. Ethanol molecules leave the MOF pores, dragging the ADC rotors into a more vertical alignment which is accompanied by significant PL enhancement. **c**, SEM images of MTF on a Si substrate (top: surface view, bottom: cross section) showing plate-like crystallites of the order of 200 nm forming a closed layer. **d**, GIWAXS diffractogram the MTF preferential $\langle 001 \rangle$ -orientation determined by excellent match with simulated diffraction peak positions (black diamonds). Laue indices are indicated for the simulated peaks that coincide with experimentally observed maxima (filled diamonds). χ is the azimuthal angle. **e**, Synthesis parameter screening. The azimuthal FWHM of the $\{112\}$ GIWAXS peak reflects the average deviation of crystallite angles from the perfect $\langle 001 \rangle$ -orientation and, therefore, provides a measure for quality of orientation. An optimum is indicated at 0.15 mM and 50 °C.

5. In the section “Oriented metal-organic framework thin films fabricated by drop-casting” in the manuscript, I think it is necessary for the authors to supplement the X-

ray photoelectron spectroscopy (XPS) and infrared spectroscopy (IR) before and after testing the loading to analyze the elemental composition and chemical valence state of the film surface, and to determine whether there is an interaction between MOF and the film based on the characteristic absorption peaks and absorption bands.

Response: We have carried out XPS and FT-IR measurements to analyze the chemical composition and the interaction between MOF and Si substrate. These revisions appear on page 3, line 34:

“The chemical composition of the MOF was characterized by Fourier-transform infrared spectroscopy (FT-IR) and X-ray photoelectron spectroscopy (XPS). The FT-IR spectra of Zn-ADC-DABCO and the ADC linker are presented in Supplementary Fig. 1. For the ADC linker, absorption bands in the 2500–3100 cm^{-1} region are assigned to the carboxylic O–H stretching vibrations. A characteristic absorption band at 1677 cm^{-1} corresponds to the C=O stretching vibration. In the spectrum of the Zn-ADC-DABCO MTF, the broad carboxylic O–H stretching vibrations (2500–3000 cm^{-1}) disappear. Concurrently, the C=O stretching vibration splits into two new bands at 1615 and 1560 cm^{-1} . These spectral changes indicate the formation of Zn–O bonds. XPS measurements confirm the presence of Zn, C, O, and N within the MTF (Supplementary Fig. 2). Furthermore, analysis of the high-resolution O 1s and N 1s XPS spectra reveals peak-fitting results reveal the coordination of these elements to zinc.”

Page 5, line 27:

“XPS analysis of the substrate interface reveals a weak peak corresponding to Si–N bonds (Supplementary Fig. 13), indicating an interaction between the Si surface and the DABCO linker.”

Supplementary Fig. 1 ATR-FTIR spectra of Zn-ADC-DABCO and ADC. The characteristic absorption bands at 1677 cm^{-1} is the stretching vibration of C=O bonds, and the absorption bands at $2500\text{--}3100\text{ cm}^{-1}$ are assigned to the carboxylic O–H stretching vibrations in curve of ADC. 1676 cm^{-1} of the stretching vibration of C=O bonds and $2500\text{--}3100\text{ cm}^{-1}$ of the carboxylic O–H stretching vibrations are disappeared in curves of Zn-ADC-DABCO.

Supplementary Fig. 2 XPS analysis of different elements. (a) Zn 2p, (b) O 1s, (c) N 1s, (d) C 1s.

Supplementary Fig. 13 XPS spectra of MTF-loaded Si and bare Si substrate. XPS analysis of the substrate interface reveals a weak peak corresponding to Si-N bonds, suggesting a chemical interaction between the Si surface and the DABCO linker.

6. As a potential nanoscale flow indicator, it is necessary to have good stability. To this end, it is recommended that the authors supplement the PXRD pattern of Zn-ADC-DABCO film placed in air and ethanol solution for 5 days.

Response: The stability of Zn-ADC-DABCO film was investigated by keeping it in air and ethanol solution for five days, respectively. The PXRD patterns of the film after these treatments are shown in Supplementary Fig. 8. Both the crystal structure and orientation remain, indicating good stability. We revised the manuscript for this stability test (see page 5 line 12):

“The stability of the Zn-ADC-DABCO film was evaluated by PXRD. As presented in Supplementary Fig. 8, the PXRD patterns of the film were collected after exposure to ambient air and immersion in ethanol for five days. The patterns confirm that the crystalline structure and orientation are retained after these treatments, indicating good stability of the MOF.”

Supplementary Fig. 8 PXRD patterns of Zn-ADC-DABCO film after kept in air and EtOH solution for 5 days. The crystal structure and orientation remain. The small diffraction peak at $\sim 28.5^\circ$ (labelled by *) stems from the mixing signals of Zn-ADC-DABCO and underlying Si substrate.

7. In the "Luminescence turn-on effect" section of the manuscript, the authors mentioned that the "flow" state caused the luminescence to turn on and off. Could this change be due to the ethanol molecules entering the MOF pores, thereby restricting the rotation of the molecular rotor, thereby causing Zn-ADC-DABCO to exhibit bright emission? Rather than the flow state? What is the difference between the "flow state" mentioned by the authors and the AIE mechanism?

Response: Thank you for this suggestion. In contrast to aggregation-induced emission (AIE)-type MOF ligands, such as diarylethenes and triphenylamines whose phenyl rings function as rotators exhibiting rotary behavior in their native state (*Adv. Sci.* 2022, 9, 2200850; *Adv. Mater.* 2025, 37, 2502742), anthracene moiety of ADC is a significantly bulkier rotator. Its rotational motion is impeded under normal conditions. Furthermore, given the small intermolecular distances within the MOF, both Fig. 2c and the time-resolved photoluminescence spectra (Fig. 4a) suggest that intermolecular interactions are the more probable cause of its luminescence question. In the light-up state, it is hypothesized that a realignment of the chromophores causes the PL enhancement. We have added respective text in the Introduction to distinguish those two mechanisms:

Page 3, line 15:

“In contrast to phenyl-based rotors, the anthracene moiety is typically regarded as a

sterically hindered rotator. Consequently, within the dense Zn-ADC-DABCO framework, its PL is quenched.³⁹ However, the application of ethanol to the MTFs triggers a dramatic change. Intriguingly, as the solvent approaches complete evaporation, the MTFs exhibit a sharp PL turn-on phenomenon. This response yields a 50-fold enhancement in intensity relative to the initial dry state. Photophysical and quartz crystal microbalance (QCM) investigations suggest that such enhancement arises from a transient realignment of MOF linkers, a process probably driven by internal stress generated during the final stage of solvent evaporation.”

Reference:

“39. Tanaka, D., *et al.* Anthracene array-type porous coordination polymer with host-guest charge transfer interactions in excited states. *Chem. Commun.* 3142-3144 (2007).”

8. In addition to the ADC linkers mentioned in this manuscript, can the authors provide films formed by similar linkers to prove the author's claim that "dynamic rearrangement of the linker achieves luminescence difference"? I think this manuscript should also provide films with molecular rotors, films without molecular rotors, and films formed by other molecular rotors for simultaneous control tests. It is far from enough to explore only one example.

Response: We appreciate your suggestion. We carried out the PL intensity evolution of Zn-NDC-DABCO with liquid ethanol evaporation. In a sense, it functions as a counterexample to the Zn-ADC-DABCO. We apologize that the current availability of linker limits us to only include Zn-NDC-DABCO. However, the investigation of novel ligands and rationally designed rotor-based MOFs will be pursued in our subsequent research. The results of Zn-NDC-DABCO are shown in Supplementary Fig. 31 and added in the revised manuscript on page 11, line 23:

“It should be noted that the simulation shows a partial rotation,²⁷ which aligns the anthracene unit more vertically, rather than a complete unhindered turnstile rotation. This simulation is in agreement with the experimental PL enhancement, because complete non-unidirectional rotation characteristically promotes non-radiative decay pathways, thereby resulting in luminescence quenching, as exemplified in the Introduction by the molecules possessing phenyl rotators.^{20,22} To verify this hypothesis, the PL evolution of Zn-NDC-DABCO was investigated during ethanol evaporation. The 1,4-naphthalenedicarboxylic acid (NDC) linker is less bulky than ADC, affording smaller steric hindrance and enough void space within the MOFs. As shown in Supplementary Fig. 31, at the transient drying stage, its luminescence is quenched. This quenching is potentially due to the less restricted rotation of the NDC linker.”

Supplementary Fig. 31 PL evolution of Zn-NDC-DABCO under dropping liquid ethanol. PL intensity kinetics of Zn-ADC-DABCO under excitation of 300 nm LED with ethanol dropping.

9. In fact, this “flow state” is related to the pores of MOF. It is recommended that the authors conduct additional testing on the pore characteristics of the Zn-ADC-DABCO film.

Response: Thank you for your suggestion. Due to the limited total mass of the MTFs deposited on the substrates, gas sorption measurements required for standard Brunauer–Emmett–Teller (BET) analysis are technically difficult. To address this challenge, we employed ethanol vapor sorption monitored via QCM to characterize the accessible porosity of the films. In addition, the good agreement observed between the experimental GIWAXS and PXRD patterns of MTFs and the simulated diffraction data derived from the single-crystal model (*Angew. Chem. Int. Ed.* 2011, 50, 8057–8061) provides strong structural evidence that the pore characterization of the MTF is consistent with that of the previously reported bulk powder analogue. The relevant discussion has been incorporated into the revised manuscript on page 8, line 11:

“Initial vapor sorption experiments were performed to establish comparison (Supplementary Fig. 24). Upon exposure to vapor, a decrease in frequency ($\Delta f/n$) is observed, indicating mass uptake. Concurrently, the bandwidth increased, indicating a rise in viscous dissipation. A stationary state was achieved within a few minutes. The values of $\Delta f/n$ exhibit slight variations across different overtones in both the dry and vapor-saturated states, reflecting the finite softness of the sample. The mass of the dry and vapor-exposed states was derived using $-\Delta f/n$ values averaged over the different

overtone. This analysis revealed a mass increase of $15\pm 2\%$ upon vapor exposure. This value is lower than the theoretical pore volume of 25% reported previously.⁵² This incomplete filling is likely attributable to the polarity mismatch between the polar ethanol guest molecules and the non-polar pores of the MOF.”

Reference:

“52. Cao, Z. J., Landström, K. N., Akhtar, F. Rapid ammonia carriers for SCR systems using MOFs $[M_2(ADC)_2(DABCO)]$ ($M = Co, Ni, Cu, Zn$). *Catalysts* **10**, 1444 (2020).”

Supplementary Fig. 24 In-situ QCM and PL measurement acquired while sample is exposed to ethanol vapor. (a) Overtone-normalized frequency shifts (b) Overtone-normalized bandwidth shifts, and (c) Evolution of PL versus time. The artifacts from the vapor adding and removing are omitted in the PL signal.

Page 3, line 43:

“The structure and preferential orientation of thin film are confirmed using GIWAXS (Fig. 1d). Excellent agreement is found between the experimentally observed diffraction peak positions and those simulated from a Zn-ADC-DABCO bulk crystal model in the $\langle 001 \rangle$ -orientation.⁴⁰”

Reference:

“40. Hirai, K., *et al.* Sequential functionalization of porous coordination polymer crystals. *Angew. Chem. Int. Edit.* **50**, 8057-8061 (2011).”

Page 4, line 3:

“In addition, the out-of-plane X-ray diffraction (PXRD) spectrum of Zn-ADC-DABCO MTF has a good agreement with the simulated diffractogram with preferred orientation along the [001] direction (Supplementary Fig. 3), indicating the film grew along the [001] direction perpendicular to the substrate surface.”

Supplementary Fig. 3 Phase structure and orientation confirmed by PXRD. PXRD pattern of Zn-ADC-DABCO (red line) and simulated diffractogram with preferred orientation along the [001] direction (black line) as reference. The small diffraction peak at $\sim 28.5^\circ$ (labelled by *) stems from the underlying Si substrate.

10. There are some minor errors in the reference format. Please check the formatting again.

Response: We thank you for your careful attention to detail. The reference formatting has been revised throughout the manuscript.

Reviewer #2 (Remarks to the Author):

This manuscript reports a solvent evaporation-induced alignment of ADC rotors within a MOF thin film, resulting in enhanced photoluminescence. The authors utilize a hot-plate drop-casting method to fabricate <001>-oriented Zn-ADC-DABCO MOF films, and employ GIWAXS, reflection spectroscopy, PL measurements, and molecular dynamics simulations to investigate the solvent-driven rotor alignment mechanism. They further propose potential applications in nanoscale fluid sensing.

While the work offers practical value in thin-film fabrication, it suffers from significant limitations in scientific novelty, mechanistic depth, and application feasibility:

Response: We thank the reviewer for assessing our manuscript. We are happy that you appreciate our approach for fabricating oriented MOF thin films (MTFs). But you give strong criticism related to the novelty, mechanism, and application feasibility. Below you can find our answers to your statements. In addition, the revisions based on your suggestions are detailed below. The revisions made in the revised manuscript are highlighted.

I. The oriented Zn-ADC-DABCO MOF film and its PL property have been well reported in other paper, which lead to a lacking novelty in this work.

Response: The authors understand that the reviewer's primary criticism is the novelty of this work. As the reviewer did not provide any specific examples, we have provided a detailed comparison with the relevant literature, specifically *Beilstein J. Nanotechnol.* 2012, 3, 570–578 and *Adv. Sci.* 2021, 8, 2001884, to clearly articulate the distinct contributions of our study. While these previous studies also report on oriented Zn-ADC-DABCO films, they rely on a layer-by-layer (LbL) deposition technique, wherein the substrate undergoes sequential immersion in alternating metal ion and linker solutions. As noted in our manuscript, the LbL method is inherently time-consuming and typically necessitates a functionalized self-assembled monolayer on the substrate to direct film growth. In sharp contrast, the drop-casting method established in this work offers a faster and facile fabrication route. Furthermore, our approach demonstrates superior versatility, allowing for deposition on a diverse range of substrates without the need for complex surface pre-treatments. To the best of our knowledge, this approach has not been previously reported. Its simplicity makes it highly attractive, offering significant potential for facilitating device integration. We believe this methodological advancement represents a significant step forward in the practical fabrication of oriented MTFs.

Regarding the PL properties, the referenced literature characterizes the PL spectra of Zn-ADC-DABCO primarily in the form of DMF-loaded nanocrystals or dried bulk powders, restricted to steady-state measurements. Conversely, our work investigates the emission properties specifically with an oriented thin film – achieving such orientation is not possible in a MOF powder and thus the opportunities for device engineering are limited. Crucially, we report the in-situ monitoring of the dynamic evolution of emission during the evaporation of ethanol (or methanol). It is well-realized that the intrinsic PL quantum yield of Zn-ADC-DABCO is extremely low (<1%) in powder forms (Chem. Commun., 2007, 3142 – 3144). However, we demonstrate that this weak emission is significantly enhanced via the solvent evaporation process. This stimuli-responsive behavior underscores the unique potential of oriented Zn-ADC-DABCO films for optical sensing applications, which are not explored in previous reports.

We recognize that these specific distinctions and the novelty of our work (methodological advancement and dynamic optical properties) may not have been sufficiently emphasized in the original submission. Therefore, we have significantly revised the manuscript to explicitly delineate these contributions and avoid any ambiguity.

Abstract:

“Sub-unit motion within metal-organic frameworks (MOFs) offers unique opportunities for nanoscale sensing. However, achieving controlled rotation of bulky linkers remains a significant challenge. In this study, a 50-fold luminescence enhancement is observed from a MOF thin film when intra-pore solvent flow orients the linker chromophores. These MOF thin films can be prepared via a facile drop-casting method on various substrates. The MOF structure consists of zinc-coordinated layers containing rotatable chromophores, separated by pillar molecules. Grazing-incidence wide-angle X-ray scattering analysis confirms the formation of highly oriented films. The deposition of a volatile organic compound, such as ethanol, triggers a significant enhancement in luminescence as the solvent nears complete evaporation. Photophysical characterization and quartz crystal microbalance measurements reveal that this phenomenon is driven by internal stress on the MOF pore level generated during the final stages of evaporation. This stress can result in a realignment of the MOF chromophores at the molecular scale. Consequently, this dynamic turn-on luminescence behavior establishes a foundation for nanoscale platforms capable of indicating solvent volatilization in real time.”

Page 3, line 4:

“To fabricate pillared-layer MTFs with preferential orientation, a straightforward drop-casting technique is employed (Fig. 1a). Typically, highly oriented pillared-layer MTFs are produced via layer-by-layer deposition, a process involving the sequential exposure of a substrate to metal and linker precursor solutions.³⁴⁻³⁶ While these conventional techniques afford control over the crystal growth of ADC-based MTFs, they are often time-consuming. Furthermore, they generally require surface functionalization with a self-assembled monolayer to support and direct oriented growth.^{37, 38} In contrast, the drop-casting method presents a simple and effective alternative, yielding a 3D pillared-layer MOF with the desirable <001>-orientation. To the best of our knowledge, this approach has not been previously reported. Its simplicity makes it highly attractive, offering significant potential for facilitating device integration.”

References:

“34. Chen, S. M., *et al.* Liquid-phase epitaxial growth of azapyrene-based chiral metal-organic framework thin films for circularly polarized luminescence. *ACS Appl. Mater. Interface* **11**, 31421-31426 (2019).

35. Ellis, J. E., Crawford, S. E., Kim, K.-J. Metal-organic framework thin films as versatile chemical sensing materials. *Mater. Adv.* **2**, 6169-6196 (2021).

36. McCarthy, B. D., *et al.* Facile orientational control of m2l2p surmofs on <100> silicon substrates and growth mechanism insights for defective mofs. *ACS Appl. Mater. Interface* **11**, 38294-38302 (2019).

37. Chandresh, A., Liu, X. J., Wöll, C., Heinke, L. Programmed molecular assembly of abrupt crystalline organic/organic heterointerfaces yielding metal-organic framework diodes with large on-off ratios. *Adv. Sci.* **8**, 2001884 (2021).

38. Zhuang, J. L., Friedel, J., Terfort, A. The oriented and patterned growth of fluorescent metal-organic frameworks onto functionalized surfaces. *Beilstein J. Nanotech.* **3**, 570-578 (2012).”

Page 3, line 15:

“In contrast to phenyl-based rotors, the anthracene moiety is typically regarded as a sterically hindered rotator. Consequently, within the dense Zn-ADC-DABCO framework, its PL is quenched.³⁹ However, the application of ethanol to the MTFs triggers a dramatic change. Intriguingly, as the solvent approaches complete evaporation, the MTFs exhibit a sharp PL turn-on phenomenon. This response yields a 50-fold enhancement in intensity relative to the initial dry state.”

Reference:

“39. Tanaka, D., *et al.* Anthracene array-type porous coordination polymer with host-guest charge transfer interactions in excited states. *Chem. Commun.* 3142-3144 (2007).”

II. The reported "flow sensing" phenomenon is based solely on solvent evaporation, representing a photophysical change rather than true dynamic fluid detection.

Response: Thank you for your comment. As already requested by Reviewer #1 (comment #3), we have revised the “flow state” in the manuscript to avoid potential ambiguity. In the revised manuscript we attribute the luminescence enhancement to internal stress exerted by the liquid ethanol upon the MTF. This is investigated by quartz crystal microbalance (QCM) with dissipation monitoring. The MTF does not act as a rigid scaffold in these experiments, instead changing while being exposed to liquid ethanol. This stress induces a microstructural deformation of the MTF during the final phase of ethanol evaporation (we also termed the "transient drying" stage), thereby precipitating a dramatic increase in luminescence intensity. Consequently, we have re-designated this condition as the "light-up" state in the revised text. The PL from the MOFs reports internal stress, which opens application perspectives. In the revised paper we have now added an in-depth discussion of the origins and potential impacts of the observed PL enhancement.

Page 8, line 8:

“To link this PL enhancement to the drying kinetics, an in-situ experimental setup combining optical measurements and QCM with dissipation monitoring was employed (Supplementary Note 2). Initial vapor sorption experiments were performed to establish comparison (Supplementary Fig. 24).”

Page 8, line 19:

“Furthermore, while a minor increase in PL is observed upon the termination of vapor exposure, the magnitude of this effect is significantly lower than the enhancement observed during liquid immersion experiments.”

Page 8, line 27:

“**Fig. 3** and Supplementary Fig. 25 illustrate the ethanol evaporation kinetics during the liquid immersion experiments. These data confirm that the immersion state itself does not yield an increase in PL. A significant PL enhancement is induced during the transient drying phase. The drying process spans ~1 minute and is characterized in the QCM data by an increase in frequency that is approximately linear with respect to time. Notably, both the frequency and bandwidth signals exhibit a temporal drift when the

layer is immersed. These sloping plateaus reflect time-dependent changes in the properties of the MTFs during the immersion state, while a behavior is absent in the vapor sorption experiments. This contrast implies that liquid ethanol induces a structural distortion within the MOF layer that vapor exposure does not. Importantly, this distortion is non-destructive and fully reversible. The reverse transformation occurs as the sample dries. It is hypothesized that this reverse transition generates transient internal stress, which drives small-scale rearrangements within the MTFs. These rearrangements effectively suppress self-quenching, thereby triggering the amplification in PL intensity observed during the transient drying stage.”

Page 9, line 19:

“Collectively, these results demonstrate that the MTFs does not function as a static rigid scaffold, its structural properties are dynamic and responsive to ethanol molecule movement inside the structure. During the transient drying stage, the evaporating solvent exerts internal stress on the framework. This stress subsequently drives a structural realignment at the molecular level, likely involving the rotation of the central anthracene moiety of the ADC linker.”

Fig. 3 Zn-ADC-DABCO thin film on molecular sieve indicates methanol flow. Shifts in overtone-normalized resonance frequency, $\Delta f/n$, and half-bandwidth, $\Delta \Gamma/n$, upon dropping 20 μL ethanol on the Quartz plate. The below schematic diagram shows the different states during ethanol evaporation.

Supplementary Fig. 25 QCM measurement. Shifts in overtone-normalized resonance frequency, $\Delta f/n$, and half-bandwidth, $\Delta\Gamma/n$, upon dropping 20 μL ethanol on the MOF thin film-coated resonator

III. The mechanism reflects an incidental conformational change triggered by hydrophobic solvent molecules, lacking a generalizable design principle or broader applicability.

Response: We wish to emphasize that this work is positioned as fundamental research. While establishing a generalizable design principle or achieving immediate broad applicability are valuable long-term objectives, the primary scope of this study is to elucidate the specific properties of this unique system. To this end, we have successfully demonstrated a proof-of-concept for the system function as an evaporation indicator. Furthermore, to ensure a balanced presentation, we have expanded the manuscript to explicitly discuss the limitations of the current approach and provide a future perspective on how these findings may be broadened in subsequent studies:

Page 11, line 38:

“Based on the finding that the PL of the MOF reports the internal stress generated during the transient drying stage, a prototypical application of Zn-ADC-DABCO as a volatile solvent evaporation indicator was demonstrated. For this proof-of-concept, the MOF was coated onto 4 Å molecular sieve beads using the drop-casting method (see Methods). When the MOF-coated sieve is positioned on a methanol droplet, liquid absorption into the sieve is initiated. Consequently, the area saturated with methanol becomes a wet region, while the area yet to absorb the liquid remains dry. Continuous solvent evaporation occurs at the interface between these two regions. It is hypothesized that the localized evaporation at this moving boundary generates internal stress, which subsequently activates the luminescence of the MOF. Indeed, the MOF-coated spherical sieve provides an eye visible optical indication of methanol evaporation (Fig. 5 and

Supplementary Video 1). In this experiment, methanol sorption and evaporation are recorded under 254 nm UV irradiation. To distinguish the position of luminescence and sorption, the upper row of Fig. 5 displays the sieve with room lighting turned off, while the lower row shows the sieve under ambient lighting. Upon initial exposure to methanol (0 s), a distinct luminescent ring emerges on the surface of the sphere. This ring migrates upward, eventually converging into a bright spot at the top. This apical emission persists for several minutes even after the exterior of the sieve appears dry (bottom Fig. 5f). The disappearance of PL suggests the cessation of solvent evaporation from the interior of the sieve. This proof-of-concept demonstrates the potential of Zn-ADC-DABCO as a functional coating for the real-time visual indication of solvent evaporation.”

Page 12, line 26:

“Although this study focused on ethanol, the underlying sensing mechanism is not inherently limited to simple alcohols. Future investigations will aim to expand the library of compatible analytes to map the operational range and limitations of these materials. Current efforts are directed toward exploring a broader spectrum of volatile organic compounds and establishing quantitative evaporation-rate dependencies via structural optimization (e.g., tuning pore size and ligand chemistry). A promising application involves integrating MOF thin films with tunable pore sizes onto the inner walls of microfluidic channels. This configuration could serve to monitor the progression of the evaporation front for various solvents and solvent mixtures. Ultimately, this work demonstrates the utility of exploiting solvent-evaporation-induced dynamic structural changes for sensing, opening new avenues for nanoscale monitoring. This approach is expected to stimulate further research into the real-time tracking of liquid volatilization through molecular-scale pores.”

Fig. 5 Zn-ADC-DABCO thin film on molecular sieve indicates methanol evaporation. Photo series of a MOF-coated molecular sieve under 254 nm excitation with room lighting off (top row) and room lighting on (bottom row). The time stamps indicate the elapsed time after exposure to the solvent. The distinctly shaped luminescence on the surface of sieve indicates

moving channels of preferred solvent evaporation from the sieve. The confined emission at the top remains for several minutes, reporting continued solvent evaporation even after the sieve appears dry already on the outside.

IV. Similar phenomena involving rotor-induced luminescence modulation in MOFs under pressure or solvent stimuli have been previously reported; this work does not constitute a substantial advancement in mechanism or functionality.

Response: Thanks for this challenge, which gives us an opportunity to better explain the novelty of our work – specifically regarding the PL modulation – by contrasting our approach with the pertinent literature cited in the manuscript. We acknowledge that rotor-induced PL modulation in MOFs has been previously reported (*Chem. Mater.* 2020, 32, 6706–672; *Adv. Mater.* 2025, 37, DOI: 10.1002/adma.202502742; *Nano-Micro Lett.* 2025, 18, DOI: 10.1007/s40820-025-01917-8). However, both the design rationale and the specific object of investigation in the present work are fundamentally distinct from these prior studies. In the cited reports, regardless of whether 1,4-benzenedicarboxylic acid or aggregation-induced emission-type ligands are employed, the functional rotator is typically a phenyl ring. Due to its small steric profile and low rotational energy barrier, the phenyl ring undergoes dynamic rotation which facilitates non-radiative decay, thereby quenching luminescence. Consequently, PL enhancement in these systems is achieved by restricting this rotation via external stimuli, such as high pressure or host-guest interactions with solvent molecules. In distinct contrast, our work employs a bulky anthracene-based linker ADC. Due to the significant steric hindrance of the anthracene unit and the limited free volume within the MTFs, the rotation of the ADC linker is already significantly impeded under ambient conditions. Unlike the phenyl-based systems, the low initial PL efficiency in our system is ascribed to strong intermolecular interactions, as referenced in *Chem. Commun.* 2007, 3142–3144. To activate emission, we utilize a solvent evaporation process that triggers a 50-fold enhancement, achieving a distinct "light-up" state. This mechanism differs fundamentally from the restriction of free rotation seen in phenyl-based rotors. Furthermore, while the referenced examples rely on MOF powders to study rotor responsiveness, our study focuses on the fabrication and characterization of oriented MOF thin films. As demonstrated in our response to comment #4, the orientation of the MTF yields a responsiveness superior to that of its powder counterpart. We have revised the manuscript to more clearly highlight these novel aspects regarding the unique rotator dynamics and the mechanism of PL enhancement.

Page 2, line 22:

“Among these, luminescent molecular rotors represent a promising class of linkers, characterized by their ability to respond to external stimuli through distinct changes in photoluminescence (PL) properties.^{20, 21} Particular attention has been paid to rotor ligands that exhibit aggregation-induced emission (AIE). For instance, Zhao *et al.* reported a series of tetraphenylethylene-based MOFs that demonstrate turn-on fluorescence in response to volatile organic compounds, temperature variations, and viscosity changes.^{22, 23} Recently, Zhu *et al.* synthesized a triphenylamine-based lanthanide MOF wherein the distortion of molecular rotors is regulated by temperature, thereby inducing multicolor luminescence switching.²⁴ In these reported systems, phenyl rings function as the rotators, a mechanism that requires only a relatively small free volume. However, incorporating bulkier units, such as anthracene, significantly increases the rotational barrier. This steric hindrance can lead to restricted rotation or even stabilize the molecule into a stationary conformation.²⁵⁻²⁷”

References:

“20. Zhang, T., *et al.* Pressure-modulated host guest interactions boost effective blue-light emission of MIL-140A nanocrystals. *Nano-Micro Lett.* **18**, DOI: 10.1007/s40820-40025-01917-40828 (2025).

21. Shustova, N. B., *et al.* Phenyl ring dynamics in a tetraphenylethylene-bridged metal-organic framework: Implications for the mechanism of aggregation-induced emission. *J. Am. Chem. Soc.* **134**, 15061-15070 (2012).

22. Dong, J. Q., *et al.* Aggregation-induced emission-responsive metal-organic frameworks. *Chem. Mater.* **32**, 6706-6720 (2020).

23. Zhang, M., *et al.* Two-dimensional metal-organic framework with wide channels and responsive turn-on fluorescence for the chemical sensing of volatile organic compounds. *J. Am. Chem. Soc.* **136**, 7241-7244 (2014).

24. Wang, H. L., *et al.* Smart lanthanide metal-organic frameworks with multicolor luminescence switching induced by the dynamic adaptive antenna effect of molecular rotors. *Adv. Mater.* **37**, DOI: 10.1002/adma.202502742 (2025).

25. Horike, S., *et al.* Dynamic motion of building blocks in porous coordination polymers. *Angew. Chem. Int. Edit.* **45**, 7226-7230 (2006).

26. Kuc, A., Enyashin, A., Seifert, G. Metal-organic frameworks: Structural, energetic, electronic, and mechanical properties. *J. Phys. Chem. B* **111**, 8179-8186 (2007).

27. Gonzalez-Nelson, A., Coudert, F. X., van der Veen, M. A. Rotational dynamics of

linkers in metal-organic frameworks. *Nanomaterials* **9**, 330 (2019).”

Page 3, line 15: “Photophysical and quartz crystal microbalance (QCM) investigations suggest that this enhancement arises from a transient reorientation inside the MOF structure, a process probably driven by internal stress generated during the final stage of solvent evaporation. This work presents a proof-of-concept demonstration of the utility of these MOFs as platforms for evaporation indicating, offering new potential for the development of advanced sensing devices.”

V. The claimed device application is overstated since the "flow state" response is transient (lasting only a few seconds), with no strategy to achieve stable or tunable signal output.

Response: We appreciate this insight. As detailed in our response to comment II, our system is designed to respond to the evaporation process rather than to function as a flow meter. We fully agree with the distinction between these two functions and have revised the manuscript to clarify this point. However, if the implication is that the evaporation kinetics are too rapid to permit effective monitoring, we respectfully disagree. In sensing applications, the primary objective is not to maintain a static or tunable output, but to effectively convert environmental stimuli into a distinguishable optical signal. In some cases, the transient nature of the response is an intrinsic characteristic of the dynamic detection mechanism. And it doesn't matter that it's transient – the response time is fast enough for us to capture as we present in the manuscript. We also emphasize that in the response to comment III, the potential of the MOF system function as indicator to monitor the solvent evaporation has been preliminarily demonstrated. We trust that the Editor can appreciate our stance here.

VI. Even beyond these major concerns , there are additional technical issues that would need to be addressed should the authors consider resubmission elsewhere

Response: We note the reviewer's comment regarding additional technical issues. While specific examples were not detailed in this particular instance, we believe we have comprehensively addressed all technical concerns in our detailed responses to the specific comments raised throughout this letter. We respectfully invite the reviewer to consider the full context of our point-by-point responses, which we believe will clarify these matters.

In addition, there are some specific suggestions and issues that need to be modified

1. phase angle (azimuthal) measurements are recommended to provide more direct and comprehensive evidence of the film orientation. This would better highlight the degree of crystallographic alignment.

Response: The texture of the Zn-ADC-DABCO MOF thin film (MTF) is characteristic of a “2D powder”. In this case, the crystallites exhibit a strong out-of-plane orientation, with their {001} crystallographic planes aligned parallel to the substrate surface, while possessing a random in-plane orientation (J. Vac. Sci. Technol. A 2001, 19, 1270–1276). 2D GIWAXS measurements performed at a fixed sample alignment with a 2D detector, are a sufficient and appropriate technique to map the reciprocal space of such textured films (Adv. Energy Mater. 2023, 13, 2300760). As shown in Fig. 1d, the 2D GIWAXS pattern presents the scattering data as a function of the out-of-plane q_z and in-plane q_r components of the scattering vector. To quantify the degree of orientation, the intensity distribution as a function of the azimuthal angle (χ , defined in Fig. 1d) was extracted for the {112} diffraction peak (Supplementary Fig. 6). From this analysis, an oriented fraction of 73% was estimated. This result confirms that the GIWAXS measurement provides sufficient and quantitative determination of the film orientation. Azimuthal angle measurement is not necessary. We have incorporated this detailed orientation analysis into the revised manuscript on page 4, line 15:

“To optimize MOF quality, the radial intensity distribution of the {112} diffraction peak was extracted as a function of the azimuthal angle χ (Fig. 1d). The degree of crystallite orientation was then evaluated by assessing the azimuthal full width at half maximum (FWHM) of this {112} diffraction peak while varying synthesis parameters, including precursor concentration, substrate temperature, and drop volume (see Supplementary Note 1 for details). Optimal crystallite orientation, characterized by the smallest FWHM (indicating minimal dispersion in sheet alignment), was achieved using a precursor concentration of 0.15 mM and a substrate temperature of 50 °C (Fig. 1e and Supplementary Figs. 4–6). For this optimized sample, the oriented fraction was estimated to be 73% via Gaussian fitting of the peak (Supplementary Fig. 7).”

Supplementary Fig. 7 Estimation of orientation degree. a) Radial intensity distribution of (112) plane as a function of azimuthal angle χ . b) Orientation degree is estimated by Gaussian peak fitting and integration of peak area.

2. The thin-film structure is confirmed only by simulated GIWAXS patterns from literature data. No single-crystal XRD or CCDC deposition is provided, limiting structural certainty. Structural refinement is recommended.

Response: As discussed in our response to the reviewer's comment #1 above, this work focuses on the synthesis and investigation of MTFs which exhibit a specific 2D powder texture. Consequently, single-crystal analysis falls outside the scope of the present study. Regarding the structural refinement, we wanted to clarify that standard Rietveld refinement requires the detection of a comprehensive set of diffraction peaks. In our case, the strong preferential orientation of the MTF results in the systematic absence or suppression of non- $\{001\}$ reflections in the standard diffraction geometry, rendering full structural refinement unfeasible. To address this, we adopted a simulation-based approach. Considering the pronounced preferential orientation, comparing the experimental data against a simulated diffraction pattern that incorporates anisotropic orientation is the appropriate method to confirm the crystalline phase (*Nat. Rev. Methods Primers* 2024, 4, 15). The crystalline real and reciprocal lattices can be rotated by applying orientation matrices. This allows the calculated reciprocal lattice points to be aligned with experimental observations, enabling rigorous qualitative phase analysis and the determination of the preferred orientation (*J. Phys. Chem. B* 2006, 110, 9882–9892). Furthermore, the experimental PXRD patterns exhibit excellent agreement with the simulations that account for the specific out-of-plane orientation. This consistency provides robust confirmation of both the crystal structure and the orientation of the Zn-ADC-DABCO MTFs. These clarifications have been incorporated into the revised manuscript on page 4, line 4:

“In addition, the out-of-plane X-ray diffraction (PXRD) spectrum of Zn-ADC-DABCO MTF has a good agreement with the simulated diffractogram with preferred orientation along the [001] direction (Supplementary Fig. 3), indicating the film grew along the [001] direction perpendicular to the substrate surface.”

Supplementary Fig. 3 Phase structure and orientation confirmed by PXRD. PXRD pattern of Zn-ADC-DABCO (red line) and simulated diffractogram with preferred orientation along the [001] direction (black line) as reference.

3. The CI-NEB simulation in Supplementary Note 3 yields a very low rotational barrier, but no temperature-dependent PL lifetime or activation energy measurements are provided for validation.

Response: We characterized the temperature-dependent PL lifetime and an activation energy of 0.514 kcal/mol was estimated by Arrhenius equation. This value is much smaller than the simulated energy barrier for ADC rotation, because the activation energy refers to the thermal quenching of PL, mostly relates to the vibration of lattice and linkers. The revision is on page 5, line 30:

“Correspondingly, the temperature-dependent fluorescence lifetime spectra revealed a low activation energy for non-radiative transitions (Supplementary Fig. 17), indicating that non-radiative processes have a significant influence on PLQY at room temperature.”

Supplementary Fig. 17 Estimation of activation energy. (a) Temperature-dependent PL lifetime spectra. (b) Activation energy estimation by Arrhenius equation $k_{nr}(T) = A \cdot \exp(-E_a/(k_B T))$, where k_{nr} is non-radiative transition rate, A is pre-exponential factor, E_a is activation energy, k_B is Boltzmann constant.

4. The observed PL enhancement during the "flow state" is based only on intensity changes, with no absolute PLQY measurement. Integrating sphere measurements are recommended.

Response: Absolute PLQY measurements using an integrating sphere were performed on dry and light-up states of Zn-ADC-DABCO MTF. These results are summarized in the Supplementary Table 1.

Supplementary Table 1 Absolute PLQY of various samples upon excitation of 365 nm.

	PLQY (%)
ADC in solution ^a	45.5±3.9 ^d
MTF ^b dry state	0.667±0.093 ^d
MTF light-up state	42.5±3.7 ^e
Powder ^c light-up state	20.0±0.9 ^e

a, $c_{ADC} = 0.1$ mM; b, MTF: Zn-ADC-DABCO thin film; c, Powder: Zn-ADC-DABCO powder deposited on quartz plate.; Statistic average from (d) twice and (e) three times.

5. The "flow state" lasts only a few seconds, which limits practical applications. Strategies to extend or control the response duration should be proposed.

Response: As detailed in our response to comment #V, the short-lasting nature of the signal does not compromise the validity of the measurement. The response kinetics are sufficiently rapid to be accurately resolved by our detection instrumentation, as

substantiated by the data presented in the manuscript. Furthermore, as emphasized in our response to Comment #III, the capability of this MOF system to function as an effective indicator for monitoring solvent evaporation has been successfully demonstrated.

6. The rotor angle change is derived entirely from simulations (49.7° vs. 30.3°). No direct experimental evidence such as solid-state NMR, EXAFS, or single-crystal XRD is provided.

Response: We thank the reviewer for this suggestion. We agree that a direct observation of the rotator angle change would provide better evidence for the proposed mechanism. However, implementing such measurements to monitor this dynamic process is technically challenging and extends beyond the instrumentation currently available to us. We have therefore designated these advanced experiments as a critical component of our future work. The objective of the present work is to establish a rapid and facile fabrication route for the oriented MTFs and to investigate its stimuli responsiveness. To this end, our GIWAXS and PXRD analyses have confirmed the crystal structure and high degree of preferential orientation of the Zn-ADC-DABCO MTF. Furthermore, the significant PL enhancement observed during ethanol evaporation demonstrates its functionality of responsiveness. Given the challenges of direct observation, we employed a suite of complementary methods to investigate the molecular rotary mechanism. These included monitoring time-resolved reflectivity and absorption, conducting time-resolved PL decay analysis, performing power-dependent PL measurements, and carrying out molecular dynamics simulations. The collective results from these varied methodologies all converge providing strong evidence that supports the proposed rotation mechanism. We are confident that this mechanism provides the most plausible explanation for our observations, and we are committed to pursuing its direct validation in future research. We trust that the Editor is convinced by the evidence we have provided.

7. Although GIWAXS after solvent cycling suggests structural retention, there is no BET or gas adsorption data to confirm pore integrity. Additional porosity characterization is recommended.

Response: Thanks for this suggestion. Due to the limited total mass of the MTFs deposited on the substrates, gas sorption measurements required for standard BET analysis are technically difficult. To address this challenge, we employed ethanol vapor sorption monitored via QCM to characterize the accessible porosity of the films. In

addition, the good agreement observed between the experimental GIWAXS and PXRD patterns of MTFs and the simulated diffraction data derived from the single-crystal model (*Angew. Chem. Int. Ed.* 2011, 50, 8057–8061) provides strong structural evidence that the pore characterization of the MTF is consistent with that of the previously reported bulk powder analogue. The relevant discussion has been incorporated into the revised manuscript on page 8, line 11:

“Initial vapor sorption experiments were performed to establish comparison (Supplementary Fig. 24). Upon exposure to vapor, a decrease in frequency ($\Delta f/n$) is observed, indicating mass uptake. Concurrently, the bandwidth increased, indicating a rise in viscous dissipation. A stationary state was achieved within a few minutes. The values of $\Delta f/n$ exhibit slight variations across different overtones in both the dry and vapor-saturated states, reflecting the finite softness of the sample. The mass of the dry and vapor-exposed states was derived using $-\Delta f/n$ values averaged over the different overtones. This analysis revealed a mass increase of $15\pm 2\%$ upon vapor exposure. This value is lower than the theoretical pore volume of 25% reported previously.⁵² This incomplete filling is likely attributable to the polarity mismatch between the polar ethanol guest molecules and the non-polar pores of the MOF.”

Reference:

“52. Cao, Z. J., Landström, K. N., Akhtar, F. Rapid ammonia carriers for SCR systems using MOFs [M₂(ADC)₂(DABCO)] (M = Co, Ni, Cu, Zn). *Catalysts* **10**, 1444 (2020).”

Supplementary Fig. 24 In-situ QCM and PL measurement acquired while sample is exposed to ethanol vapor. (a) Overtone-normalized frequency shifts (b) Overtone-normalized bandwidth shifts, and (c) Evolution of PL versus time. The artifacts from the vapor adding and removing are omitted in the PL signal.

Page 3, line 43:

“The structure and preferential orientation of thin film are confirmed using GIWAXS (Fig. 1d). Excellent agreement is found between the experimentally observed diffraction peak positions and those simulated from a Zn-ADC-DABCO bulk crystal model in the <001>-orientation.⁴⁰”

Reference:

“40.Hirai, K., *et al.* Sequential functionalization of porous coordination polymer crystals. *Angew. Chem. Int. Edit.* **50**, 8057-8061 (2011).”

Fig. 1d, GIWAXS diffractogram the MTF preferential <001>-orientation determined by excellent match with simulated diffraction peak positions (black diamonds). Laue indices are indicated for the simulated peaks that coincide with experimentally observed maxima (filled diamonds). χ is the azimuthal angle.

In conclusion, I do not recommend this manuscript for publication in *Nature Communications*.

We thank the reviewer for their time and constructive evaluation of our manuscript. We believe we have now thoroughly addressed all the concerns raised and that the manuscript has been substantially improved. We have done our best and have invested a huge effort in addition experimentation and analysis, and hope the revised manuscript is now suitable for publication in *Nature Communications*. We trust that the Editor agrees here.

Reviewer #3 (Remarks to the Author):

This manuscript presents an investigation of a MOF used as a photoluminescent (PL) probe during solvent evaporation. The authors describe a simple drop-casting procedure to fabricate oriented films of this MOF, which are then employed to monitor PL intensity changes during evaporation. They attribute the observed increase in PL to a flow-induced alignment of molecular rotors.

The work is overall interesting, particularly in its attempt to correlate evaporative flow with optical changes in a MOF. However, several critical aspects require clarification and evidences for instance on the relevance of this system for real applications. The key claims, especially regarding the sensing mechanism and the MOF orientation process, are not sufficiently supported by experimental data. Below I provide detailed comments and suggestions for improvement:

We appreciate the reviewer for the careful reading of our manuscript and the positive appraisal along with constructive suggestions. More experimental results to support the mechanisms and the MOF orientation are provided in the revised paper. The revisions made in the manuscript are **highlighted**.

1) One of the main issue is related to the relevance of this approach. Beyond the interesting observation, the practical relevance of the proposed approach is not clearly justified. Beyond ethanol (and possibly methanol), it is unclear how this could translate into real applications. What would be the envisioned device? For which solvents? A more in-depth discussion on potential use cases or limitations would be valuable.

Response: We thank the reviewer for raising this important point. We have added a discussion to the revised manuscript. The added text reads (page 12, line 26):

“Although this study focused on ethanol, the underlying sensing mechanism is not inherently limited to simple alcohol. Future investigations will aim to expand the library of compatible analytes to map the operational range and limitations of these materials. Current efforts are directed toward exploring a broader spectrum of volatile organic compounds and establishing quantitative evaporation-rate dependencies via structural optimization (e.g., tuning pore size and ligand chemistry). A promising application involves integrating MOF thin films with tunable pore sizes onto the inner walls of microfluidic channels. This configuration could serve to monitor the progression of the evaporation front for various solvents and solvent mixtures. Ultimately, this work demonstrates the utility of exploiting solvent-evaporation-induced dynamic structural changes for sensing, opening new avenues for nanoscale monitoring. This approach is expected to stimulate further research into the real-time tracking of liquid volatilization

through molecular-scale pores.”

2) On the oriented MOF: the ability to obtain oriented Zn-ADC-DABCO films by simple drop casting is a potentially impactful claim. Orientation in MOFs is typically non-trivial and often requires external stimuli or epitaxial growth. Two mechanisms are discussed in SI Figure S6. Mechanism (b) appears more plausible, especially since orientation occurs on various substrates. Is this driven by sedimentation, or by capillary forces during drying? Could pre-crystallization in solution followed by controlled evaporation play a role? In this case this would resemble to other previously reported approaches (Mater. Chem. Front., 2023,7, 5545-5560) Experiments that address these points are necessary to support this claim.

Response: We thank the reviewer for this insightful suggestion. We have conducted investigations into pre-crystallization by changing two parameters: 1) the aging duration of the precursor solution prior to drop-casting and 2) the ethanol evaporation duration during the MOF thin films (MTFs) formation process. For the aging study, a duration of 0 minutes signifies that the solution was deposited on the substrate immediately following the admixture of the Zn ions and linkers. Regarding solvent evaporation, a 30 μL droplet of the precursor solution was observed to reach complete dry in ~ 30 seconds at 50 $^{\circ}\text{C}$. This evaporation period was extended to 80 s and 200 s by covering the hot plate. Both approaches are hypothesized to increase the degree of pre-crystallization in the solution. The results indicate that a longer duration correlates with a deterioration in the preferential orientation of the resulting MTFs (Supplementary Fig. 12). This finding suggests that the synthesis mechanism is not governed solely by the sedimentation of pre-formed crystal grains.

XPS analysis of the substrate interface reveals a weak peak corresponding to Si-N bonds (Supplementary Fig. 13), suggesting a chemical interaction between the Si surface and the DABCO linker. Furthermore, we employed the Scherrer equation to estimate the crystallite thickness (distinct from the overall MTF thickness). If a sequentially epitaxial growth of crystallites occurred with successive deposition cycles, the crystallite size would be expected to increase with each deposition step. However, our analysis revealed that although the total volume of the drop-cast solution was increased from 200 μL to 1600 μL (applied in 30 μL per droplet), the estimated crystallite size remained constant (Supplementary Fig. 14). This observation rules out a mechanism predicated on stepwise epitaxial surface growth.

Based on these results, we propose that the oriented growth of the MTF, particularly at the droplet periphery, is governed by a combination of two mechanisms

(Supplementary Fig. 15): 1) crystallization from the surface via the formation of a pillar molecule functional layer on a metal-oxide substrate; 2) formation of nanoplatelets in solution adopts a lying-flat morphology on the substrate due to minimization of surface energy and horizontal capillary forces during ethanol evaporation. These revisions and the accompanying discussion have been incorporated into the manuscript on page 5, line 22:

“To elucidate the formation mechanism of the oriented MOF, GIWAXS of MTF prepared from pre-crystallized solutions, crystallite thickness of MTF made with various solution volumes, and XPS of the MTF-loaded Si substrate interface were studied. As shown in Supplementary Fig. 12, an increased degree of pre-crystallization results in poor MTF orientation. This suggests that the sedimentation of pre-formed crystal grains is not the governing mechanism. XPS analysis of the substrate interface reveals a weak peak corresponding to Si–N bonds (Supplementary Fig. 13), indicating an interaction between the Si surface and the DABCO linker. However, the observation of a constant crystallite size, regardless of increasing solution volumes (Supplementary Fig. 14), rules out a stepwise epitaxial surface growth mechanism. Based on these results, the oriented growth of the MOF, particularly at the droplet periphery, is likely governed by a combination of two mechanisms (discussed further in Supplementary Fig. 15): 1) surface-initiated crystallization, mediated by a functional layer of pillar molecules on the substrate; 2) formation of nanoplatelets in solution, which subsequently adopt a lying-flat morphology on the substrate, driven by surface energy minimization and horizontal capillary forces during ethanol evaporation.^{41,42}”

References:

“41. Park, J., Moon, H. R., Kim, J. Y. Macroscopic alignment of metal-organic framework crystals in specific crystallographic orientations. *Mater Chem Front* **7**, 5545-5560 (2023).

42. Fan, H. C., *et al.* Methylamine-assisted secondary grain growth for CH₃NH₃PbI₃ perovskite films with large grains and a highly preferred orientation. *J Mater Chem A* **9**, 7625-7630 (2021).”

Supplementary Fig. 12 FWHM of χ obtained from MTF made by pre-crystallization solution. a) Aging duration of the precursor solution prior to drop-casting. b) Ethanol evaporation duration during the MTF formation process.

Supplementary Fig. 13 XPS spectra of MTF-loaded Si and bare Si substrate. XPS analysis of the substrate interface reveals a weak peak corresponding to Si-N bonds, suggesting a chemical interaction between the Si surface and the DABCO linker.

Supplementary Fig. 14 Crystallite thickness over drop-cast growth solution volume. The estimated crystallite size (thickness) is based on the Scherrer equation ($d = K\lambda/(\beta\cos\theta)$), with the Scherrer constant $K \approx 0.9$, the wavelength of X-ray photons $\lambda \approx$

0.154 nm, and the diffraction angle θ) and the FWHM (β , in radians) of the first diffraction peak progression of the PXRD patterns in Supplementary Fig. 5.

Supplementary Fig. 15 Possible growth mechanisms leading to preferred MOF thin film orientation during drop-casting on hot plate. 1) MOF thin film growth by crystallization from the surface, via the formation of a pillar molecule functional layer on a metal-oxide substrate. 2) MOF thin film growth by formation of nanoplatelets in solution and adopts lying-flat morphology driving by lowest-surface-energy facet and horizontal capillary force. A plate-like crystallize shape, thereby, favors ordered stacking on the substrate surface. As shown by Hupp and co-workers,¹⁸ DABCO molecules can directly interact with a metal-oxide surface enabling surface-initiated MOF film growth with the pillar molecules vertically oriented. Secondly, literature suggests that the bond between Zn paddle wheels and pillar molecules can be attacked by water.^{19, 20} However, steric protection by bulky groups of surrounding linkers, such as ADC, can prevent a pillar exchange for water.^{20, 21} Based on this mechanism, the formation of the Zn-ADC paddle wheels is a necessary prerequisite for the stable installation of DABCO pillars. Consequently, the MOF exhibits a faster growth rate along the layer linker plane, resulting in MOF platelets which are most likely to lie flat on the substrate. This is also an explanation for the indication of preferred orientation on the silver surface (Supplementary Fig. 10e), which does not offer the possibility of DABCO-anchoring. We suggest that the relative stability of the Zn-ADC paddle wheel is key to allowing this facile fabrication process. Testing the synthesis protocol using successively less bulky linkers 1,4-naphthalene-dicarboxylic acid (NDC) and 1,4-benzene-dicarboxylic acid (BDC) still seems to produce crystalline structures. However, we observe a significant loss of preferred orientation and long-range atomic order the smaller the aromatic central group (Supplementary Fig. 16). Further searching and targeted synthesis design could extend the number of linkers that function with this highly attractive and easily scalable fabrication process.

3) page 3 The authors assess orientation via the FWHM of the {112} diffraction peak. However, changes in synthesis conditions likely affect crystal size, not just orientation. The crystal size should be characterized. A more robust method could involve comparing intensity ratios of multiple peaks.

Response: Your comment is appreciated. We wanted to clarify that we assess the degree

of orientation via the FWHM of the intensity distribution as a function of the **azimuthal angle** (χ , defined in Fig. 1d) for the {112} diffraction peak. This method is distinct from analyzing the FWHM of the peak in q value. Operationally, this is achieved by integrating the intensity of the {112} Bragg peak along the azimuthal χ direction. The azimuthal FWHM is then obtained by a Gaussian fit of this distribution. Consequently, a smaller azimuthal FWHM value corresponds to a narrower orientation distribution and thus a higher degree of preferred orientation. We have revised the corresponding description of this method in the manuscript on page 5 line 3:

“To optimize MOF quality, the radial intensity distribution of the {112} diffraction peak was extracted as a function of the azimuthal angle χ (Fig. 1d). The degree of crystallite orientation was then evaluated by assessing the azimuthal full width at half maximum (FWHM) of this {112} diffraction peak while varying synthesis parameters, including precursor concentration, substrate temperature, and drop volume (see Supplementary Note 1 for details). Optimal crystallite orientation, characterized by the smallest FWHM (indicating minimal dispersion in sheet alignment), was achieved using a precursor concentration of 0.15 mM and a substrate temperature of 50 °C (Fig. 1e and Supplementary Figs. 4–6). For this optimized sample, the oriented fraction was estimated to be 73% via Gaussian fitting of the peak (Supplementary Fig. 7).”

Supplementary Fig. 7 Estimation of orientation degree. a) Radial intensity distribution of (112) plane as a function of azimuthal angle χ . b) Orientation degree is estimated by Gaussian peak fitting and integration of peak area.

We agree that comparing the intensity ratios of multiple diffraction peaks is a valid alternative strategy for assessing orientation. However, as shown in Supplementary Fig. 3, the PXRD diffractograms only display the {001} series of reflections (i.e., (002), (004), and (006)). While this strongly confirms the high degree of out-of-plane orientation, it simultaneously precludes the use of intensity ratios of different peaks from other crystallographic planes in this measurement geometry. As for the crystal size

characterization, we have mentioned the thickness of crystallites in response of comment #2.

Supplementary Fig. 3 Phase structure and orientation confirmed by PXRD. PXRD pattern of Zn-ADC-DABCO (red line) and simulated diffractogram with preferred orientation along the [001] direction (black line) as reference. The small diffraction peak at $\sim 28.5^\circ$ (labelled by *) stems from the underlying Si substrate.

4) The porosity of the MOF film is not quantified, yet it is central to the flow and PL mechanism proposed. What is the pore size distribution, porous volume % and the interparticle void fraction in the film? Since scattering is later claimed to be negligible, techniques like ellipsometric porosimetry or krypton adsorption on thin films could be employed to provide these informations.

Response: Thanks for your suggestion. Due to the limited total mass of the MTFs deposited on the substrates, gas sorption measurements required for standard Brunauer–Emmett–Teller (BET) analysis are technically difficult. To address this challenge, we employed ethanol vapor sorption monitored via QCM to characterize the accessible porosity of the films. In addition, the good agreement observed between the experimental GIWAXS and PXRD patterns of MTFs and the simulated diffraction data derived from the single-crystal model (*Angew. Chem. Int. Ed.* 2011, 50, 8057–8061) provides strong structural evidence that the pore characterization of the MTF is consistent with that of the previously reported bulk powder analogue. The relevant discussion has been incorporated into the revised manuscript on page 8, line 11:

“Initial vapor sorption experiments were performed to establish comparison (Supplementary Fig. 24). Upon exposure to vapor, a decrease in frequency ($\Delta f/n$) is observed, indicating mass uptake. Concurrently, the bandwidth increased, indicating a rise in viscous dissipation. A stationary state was achieved within a few minutes. The values of $\Delta f/n$ exhibit slight variations across different overtones in both the dry and vapor-saturated states, reflecting the finite softness of the sample. The mass of the dry and vapor-exposed states was derived using $-\Delta f/n$ values averaged over the different overtones. This analysis revealed a mass increase of $15 \pm 2\%$ upon vapor exposure. This value is lower than the theoretical pore volume of 25% reported previously.⁵² This incomplete filling is likely attributable to the polarity mismatch between the polar ethanol guest molecules and the non-polar pores of the MOF.”

Reference:

“52. Cao, Z. J., Landström, K. N., Akhtar, F. Rapid ammonia carriers for SCR systems using MOFs [M₂(ADC)₂(DABCO)] (M = Co, Ni, Cu, Zn). *Catalysts* **10**, 1444 (2020).”

Supplementary Fig. 24 In-situ QCM and PL measurement acquired while sample is exposed to ethanol vapor. (a) Overtone-normalized frequency shifts (b) Overtone-normalized bandwidth shifts, and (c) Evolution of PL versus time. The artifacts from the vapor adding and removing are omitted in the PL signal.

Page 3, line 43:

“The structure and preferential orientation of thin film are confirmed using GIWAXS (Fig. 1d). Excellent agreement is found between the experimentally observed

diffraction peak positions and those simulated from a Zn-ADC-DABCO bulk crystal model in the $\langle 001 \rangle$ -orientation.⁴⁰

Reference:

“40.Hirai, K., *et al.* Sequential functionalization of porous coordination polymer crystals. *Angew. Chem. Int. Edit.* **50**, 8057-8061 (2011).”

Fig. 1d, GIWAXS diffractogram the MTF preferential $\langle 001 \rangle$ -orientation determined by excellent match with simulated diffraction peak positions (black diamonds). Laue indices are indicated for the simulated peaks that coincide with experimentally observed maxima (filled diamonds). χ is the azimuthal angle.

Page 4, line 3:

“In addition, the out-of-plane X-ray diffraction (PXRD) spectrum of Zn-ADC-DABCO MTF has a good agreement with the simulated diffractogram with preferred orientation along the $[001]$ direction (Supplementary Fig. 3), indicating the film grew along the $[001]$ direction perpendicular to the substrate surface.”

5) Figure 2 No comparison is shown with non-oriented or powder-based MOF films during solvent evaporation. This is essential to confirm either that the PL enhancement arises from orientation, and not simply from solvent removal or to justify the use of oriented MOFs. Similarly, the reflectivity experiments later in the manuscript would benefit from such control samples.

Response: Thank you for your suggestion. We compared the PL behaviors of oriented MTFs and powder-deposited MOF film. In addition, the evolution of absorption of these two samples is monitored during evaporation. We have added the PL and absorption results in the revised manuscript and SI. The text reads (page 8, line 12):

“For comparison, the evolution of PL and PLQY was studied for Zn-ADC-DABCO

powder samples deposited on substrates. As presented in Supplementary Fig. 20 and Table 1, the oriented MTF exhibits a superior luminescence enhancement in the light-up state compared to the powder counterpart. This finding highlights the role of crystallite orientation in governing the responsiveness to ethanol evaporation, suggesting the behavior of a more vertical tilt of the anthracene rotators at light-up state.”

Page 10 line 12:

“Furthermore, the evolution of absorption was studied for both the oriented Zn-ADC-DABCO MTF and its powder deposited on a quartz plate. As depicted in Supplementary Fig. 29, the oriented MTF exhibits a more significant decrease in absorbance than its powder counterpart. These findings suggest the rotation of ADC linkers towards a perpendicular orientation within the oriented MTF, which is in agreement with the conclusions previously derived from the PL behavior.”

Supplementary Fig. 20 Comparison on PL of oriented MTF and MOF powder deposited on Si substrate. PL intensity (averaged from 410 to 430 nm) over time.

Supplementary Fig. 29 Comparison on absorption of oriented MTF and MOF powder deposited on quartz plate. Absorbance intensity at 365 nm over time.

Supplementary Table 1 Absolute PLQY of various samples upon excitation of 365 nm.

	PLQY (%)
ADC in solution ^a	45.5±3.9 ^d
MTF ^b dry state	0.667±0.093 ^d
MTF light-up state	42.5±3.7 ^e
Powder ^c light-up state	20.0±0.9 ^e

a, $c_{\text{ADC}} = 0.1 \text{ mM}$; b, MTF: Zn-ADC-DABCO thin film; c, Powder: Zn-ADC-DABCO powder deposited on quartz plate.; Statistic average from (d) twice and (e) three times.

6) Page 5 The observation that larger droplet volumes yield higher PL intensity during the "flow state" is inconsistent with the proposed mechanism. Assuming a fluid flow inside pores, the amount of ethanol entering the pore system should not scale with the droplet volume. At most, larger droplets should extend the duration of the evaporation process, not increase the amplitude of the PL response (the number of molecular rotors is the same). Please clarify or reconsider this interpretation.

Response: Thank you for your comment. As we answer Reviewer 1's comment #3, we have decided to revise the term "flow state" to avoid ambiguity, according to the comments of all reviewers and our re-evaluation. In the revised manuscript, we attribute the luminescence enhancement to internal stress exerted by the liquid ethanol upon the MTF. This is investigated by quartz crystal microbalance (QCM) with dissipation monitoring (Please see detailed revision in comment #8). This stress induces a microstructural deformation of the MTF during the final phase of ethanol evaporation (we also termed the "transient drying" stage), thereby precipitating a dramatic increase in luminescence intensity. Consequently, we have re-designated this condition as the "light-up" state in the revised text. Here, the higher PL intensity with volume of ethanol droplet is due to the increased internal stress with longer duration of liquid ethanol. The detailed text reads (page 9, line 9):

“To further validate this mechanism, varying volumes of liquid ethanol were applied to the film. As illustrated in Supplementary Fig. 26, the magnitude of PL enhancement increases with the volume of ethanol, a trend consistent with the results shown in Fig. 2e. QCM result reveals that larger droplet volumes prolong the immersion duration before the MTF dries. Extended exposure to the liquid ethanol correlates with a more pronounced drift in the QCM frequency and bandwidth signals, indicating more extensive structural perturbations within the film. Consequently, the stress generated during the transient drying stage increases or increasingly spreads across the whole MTF, resulting in a higher degree of PL enhancement.”

Supplementary Fig. 26. QCM and PL measurements with different volumes of liquid ethanol. Top: Shifts in overtone-normalized resonance frequency, $\Delta f/n$, and half-bandwidth, $\Delta\Gamma/n$, upon dropping 20 μL (left) and 10 μL (right) ethanol on the MOF thin film-coated resonator. Bottom: PL measurement is performed in parallel.

7) Page 6 The interpretation of reflectivity changes as purely due to reduced absorption is not fully convincing. Interferometric effects due to changes in refractive index or film thickness (or even thin liquid layers forming on top) may strongly contribute: the reflectivity modification due to interferences is most pronounced at shorter wavelengths. Please provide full reflectance spectra and discuss these alternative explanations. How is the non-flat liquid interface during evaporation handled for reflectivity? also for experiments on Si substrates? Absorption spectroscopy on transparent substrates could help resolve this. And in any case, reference experiment with not oriented MOF should be provided to support the claim.

Response: The evolution of reflection signal on Si substrate during EtOH evaporation is shown in Supplementary Fig. 28. We apologize for the full reflectance spectra that is impossible to collect due to the instrument limitation. However, as response in comment #5, the absorption behaviors of oriented MTF and powder-based film are compared. The corresponding revision is highlighted in the manuscript and shown in the response to comment #5.

Supplementary Fig. 28 Reflection of Si substrate at 260 nm during ethanol evaporation. In contrast to the Zn-ADC-DABCO thin film, there is only the sudden transition from wet to dry state visible and no increase of the reflection signal right before. The reflection curves at 300 and 500 nm in Fig. 4c closely resemble this step-like behavior from wet to dry state.

8) The discussion on internal flows is currently a little simplified. While flow during evaporation can occur, microporous materials like MOFs can also exhibit vapor-phase transport (e.g., Knudsen diffusion) rather than liquid capillary flow, depending on the pore size, connectivity, and degree of saturation. To probe that, I recommend to perform sorption analysis on the films (ellipsometry porosimetry for instance) in presence of EtOH vapors to provide EtOH desorption isotherms and eventually thickness contraction during desorption. For instance, during evaporation huge capillary force due the formation of a meniscus into the pores can also play a role on the modification of molecules alignment.

Response: Thank you for your suggestion. As response presented in comment #6, we have revised the mechanism. The revised content is added in the manuscript on page 8, line 8:

“To link this PL enhancement to the drying kinetics, an in-situ experimental setup combining optical measurements and QCM with dissipation monitoring was employed (Supplementary Note 2). Initial vapor sorption experiments were performed to establish comparison (Supplementary Fig. 24).”

Page 8, line 19:

“Furthermore, while a minor increase in PL is observed upon the termination of vapor exposure, the magnitude of this effect is significantly lower than the enhancement

observed during liquid immersion experiments.”

Page 8, line 27:

“**Fig. 3** and Supplementary Fig. 25 illustrate the ethanol evaporation kinetics during the liquid immersion experiments. These data confirm that the immersion state itself does not yield an increase in PL. A significant PL enhancement is induced during the transient drying phase. The drying process spans ~1 minute and is characterized in the QCM data by an increase in frequency that is approximately linear with respect to time. Notably, both the frequency and bandwidth signals exhibit a temporal drift when the layer is immersed. These sloping plateaus reflect time-dependent changes in the properties of the MTFs during the immersion state, while a behavior is absent in the vapor sorption experiments. This contrast implies that liquid ethanol induces a structural distortion within the MOF layer that vapor exposure does not. Importantly, this distortion is non-destructive and fully reversible. The reverse transformation occurs as the sample dries. It is hypothesized that this reverse transition generates transient internal stress, which drives small-scale rearrangements within the MTFs. These rearrangements effectively suppress self-quenching, thereby triggering the amplification in PL intensity observed during the transient drying stage.”

Page 9, line 19:

“Collectively, these results demonstrate that the MTFs does not function as a static rigid scaffold, its structural properties are dynamic and responsive to ethanol molecule movement inside the structure. During the transient drying stage, the evaporating solvent exerts internal stress on the framework. This stress subsequently drives a structural realignment at the molecular level, likely involving the rotation of the central anthracene moiety of the ADC linker.”

Fig. 3 Zn-ADC-DABCO thin film on molecular sieve indicates methanol flow. Shifts in overtone-normalized resonance frequency, $\Delta f/n$, and half-bandwidth, $\Delta\Gamma/n$, upon dropping 20 μL ethanol on the Quartz plate. The below schematic diagram shows the different states during ethanol evaporation.

Supplementary Fig. 25 QCM measurement. Shifts in overtone-normalized resonance frequency, $\Delta f/n$, and half-bandwidth, $\Delta\Gamma/n$, upon dropping 20 μL ethanol on the MOF thin film-coated resonator

9) the experiment with the spherical molecular sieves should be better described and justified. It is not clear (to me at least) where the PL MOFs is located and why the luminescence is observed only on the top. Why this system was chosen as proof of concept? I'm still unconvinced about its relevance. As also in my comment 1, I suggest clarifying this point or provide a more relevant example.

Response: We appreciate your suggestion. We have revised this part about MOF-coated sieve on page 11, line 38:

“Based on the finding that the PL of the MOF reports the internal stress generated

during the transient drying stage, a prototypical application of Zn-ADC-DABCO as a volatile solvent evaporation indicator was demonstrated. For this proof-of-concept, the MOF was coated onto 4 Å molecular sieve beads using the drop-casting method (see Methods). When the MOF-coated sieve is positioned on a methanol droplet, liquid absorption into the sieve is initiated. Consequently, the area saturated with methanol becomes a wet region, while the area yet to absorb the liquid remains dry. Continuous solvent evaporation occurs at the interface between these two regions. It is hypothesized that the localized evaporation at this moving boundary generates internal stress, which subsequently activates the luminescence of the MOF. Indeed, the MOF-coated spherical sieve provides an eye visible optical indication of methanol evaporation (Fig. 5 and Supplementary Video 1). In this experiment, methanol sorption and evaporation are recorded under 254 nm UV irradiation. To distinguish the position of luminescence and sorption, the upper row of Fig. 5 displays the sieve with room lighting turned off, while the lower row shows the sieve under ambient lighting. Upon initial exposure to methanol (0 s), a distinct luminescent ring emerges on the surface of the sphere. This ring migrates upward, eventually converging into a bright spot at the top. This apical emission persists for several minutes even after the exterior of the sieve appears dry (bottom Fig. 5f). The disappearance of PL suggests the cessation of solvent evaporation from the interior of the sieve. This proof-of-concept demonstrates the potential of Zn-ADC-DABCO as a functional coating for the real-time visual indication of solvent evaporation.”

Fig. 5 Zn-ADC-DABCO thin film on molecular sieve indicates methanol evaporation. Photo series of a MOF-coated molecular sieve under 254 nm excitation with room lighting off (top row) and room lighting on (bottom row). The time stamps indicate the elapsed time after exposure to the solvent. The distinctly shaped luminescence on the surface of sieve indicates moving channels of preferred solvent evaporation from the sieve. The confined emission at the top remains for several minutes, reporting continued solvent evaporation even after the sieve appears dry already on the outside.

RESPONSE TO REVIEWERS' COMMENTS

Reviewer #1 (Remarks to the Author):

I am satisfied with the author's revisions. Agree to accept and publish.

We thank the reviewer for the appreciation of our work and the insightful comments given.

Reviewer #2 (Remarks to the Author):

I think the authors have addressed the review comments.

We appreciate the reviewer for the recognition of our work and the great help in the improvement of the manuscript.

Reviewer #3 (Remarks to the Author):

I have carefully analyzed the revised version of the manuscript as well as the authors' responses to the reviewers. I appreciate the authors' efforts to revise the manuscript and to shift the focus toward the development of evaporative sensors rather than flow sensors. However, several important concerns remain insufficiently addressed, and some responses are unsatisfactory and I am still not convinced of its relevance or of some of the main scientific claims.

We appreciate you for the insightful suggestions. We have revised the figures displaying the combined QCM and PL data to more clearly demonstrate the underlying mechanism. Additionally, we have performed reflectance measurements. The revisions made in the manuscript are highlighted.

More specifically:

1. point 1 on relevance. The added paragraph regarding potential applications remains overly generic and does not convincingly demonstrate the broader relevance of the device beyond ethanol evaporation detection (is it very relevant problem?). As it stands, the justification and versatility of the method appears speculative and lacks concrete examples, which weakens the argument for publication in Nature Communications.

Response: We appreciate the reviewer's perspective regarding the broader impact of this work. We acknowledge that demonstrating a broader range of device applications would enhance practical significance, and we have successfully demonstrated a proof-

of-concept for this system functioning as a real-time evaporation indicator (**Section: Proof-of-concept evaporation indicator**). However, the engineering of such concrete devices represents a long-term objective that lies beyond the current scope of this fundamental investigation.

We wish to clarify that the primary objective of this study is fundamental research: to elucidate the fabrication of oriented MOF thin films (MTFs) and their distinct turn-on photophysical mechanism responding to external stimuli. Typically, the MOF thin film exhibits negligible PL in the dry and slightly increased PL in the immersion state. Strong PL is exclusively observed during the kinetic transition from the wet to the dry state. In dry state, the chromophores adopt a packing arrangement that facilitates intermolecular coupling and subsequent PL quenching. However, when the chromophores are forced into a non-equilibrium transition state during solvent evaporation, i.e. rotation of ADC linkers, this inter-chromophore coupling is disrupted. Consequently, the quenching pathway is suppressed, and strong PL is transiently activated. This phenomenon of transient PL modulated by dynamic molecular motion is unprecedented and of significant fundamental interest. This mechanism transcends the specific context of solvent evaporation, suggesting a novel principle for stimuli-responsive materials. For instance, guest transport within the MTF pores may induce rotor rearrangement and change of internal molecular strain, paving the way for the development of nanoscopic flow sensors. Consequently, we believe these findings hold broad relevance that aligns well with the multidisciplinary readership of Nature Communications.

To ensure a balanced presentation, we have revised the manuscript to discuss the boundaries of the current approach and provide a future perspective on how these fundamental findings may be translated into broader applications in subsequent studies. The text reads on page 12, line 35:

“The ability of molecular rotors to signal conformational transitions establishes a foundation for the development of nanoscale evaporation indicators. Moreover, the underlying sensing mechanism based on molecular conformational changes has implications extending well beyond solvent evaporation. For instance, guest transport within the MTF pores may induce change of internal molecular strain and rotor rearrangement, paving the way for the development of nanoscopic flow sensors. Future investigations should aim to expand the library of compatible analytes and establish quantitative stimuli-response dependencies via structural optimization (e.g., tuning pore size and ligand chemistry). Ultimately, this work is expected to stimulate further research into the real-time tracking of volatilization and fluid transport through molecular-scale pores.”

2. Point 4 on the porosity.

The QCM measurements indicate adsorption and desorption of ethanol; however, they do not provide meaningful insight into pore size distribution or interparticle porosity, that as I indicated in my initial review, both of which may strongly influence the photoluminescence (PL) response during evaporation. Presenting the data in the form of adsorption isotherms may clarify this point and help determine whether porosity affects the sensing mechanism.

Response: Thank you for this suggestion. We acknowledge that QCM-based ethanol sorption measurements do not yield pore size distribution or interparticle porosity data. As noted in our previous response, due to the limited total mass of the Zn-ADC-DABCO thin films, it is not feasible for us to conduct reliable adsorption isotherms for BET analysis. Standard volumetric gas sorption techniques require sample quantities that significantly exceed the mass available in these thin films. However, QCM analysis is widely accepted within the MOF thin-film community as the effective standard for quantifying guest molecule uptake and verifying porosity (*J. Phys. D: Appl. Phys.* 2017, 50, 193004; *Phys. Chem. Chem. Phys.* 2013, 15, 9295-9299). Our QCM data explicitly monitors the uptake of ethanol molecules ($15\pm 2\%$), providing evidence of accessible pores within the Zn-ADC-DABCO thin films.

Furthermore, the porosity of our samples is supported by their high crystallographic consistency to the known bulk phase. The GIWAXS data and the derived pseudo-XRD pattern (**Fig. R1a**, obtained via radial integration) exhibit excellent agreement with the simulated diffractogram of the reported Zn-ADC-DABCO structure (CCDC: 814078, *Angew. Chem. Int. Ed.* 2011, 50, 8057–8061). As visualized in the crystallographic models presented in **Fig. R1b–c**, this framework possesses intrinsic lattice porosity, confirming that the pore architecture is preserved in our thin-film samples. Moreover, a previous study has successfully employed this identical single-crystal model (CCDC: 814078) to refine the structure of bulk Zn-ADC-DABCO powder (*Catalysts* 2020, 10, 1444), reporting a BET surface area of $640\text{ m}^2/\text{g}$, a micropore area of $440\text{ m}^2/\text{g}$, and a micropore volume of $0.23\text{ cm}^3/\text{g}$. Given that our GIWAXS data confirm that our thin films have same crystal structure as this powder, it is reasonable to infer that they possess comparable intrinsic porosity. While the low total mass of the thin films precludes direct pore size distribution analysis via conventional gas sorption, the convergence of QCM data, structural validation (GIWAXS), and the strong agreement with established literature values collectively verify the porous nature of our Zn-ADC-DABCO films.

Fig. R1 (a) Pseudo-XRD obtained by radial integration of 2D GIWAXS data over all azimuthal data. (b and c) Sketch of the partial MOF structure with the distance between the carbons and the distance between the hydrogens view along the (b) [110] and (c) [001] directions.

3. point 6 on Mechanism of PL Enhancement. The new experiment does not satisfactorily address my earlier concern. If the proposed mechanism is correct—namely, that the PL enhancement originates from internal stresses generated during evaporation—then the PL intensity should be independent of droplet volume. The magnitude of internal stress at a given moment should not scale with the total amount of evaporating liquid. The results shown in Figure S26 do not support the proposed mechanism and therefore leave a core claim unvalidated.

Response: We appreciate your assessment regarding the dependence of PL on droplet volume. We agree that the droplet volume does not directly influence the PL intensity. Rather, it is governed by the duration of solvent exposure (immersion time), which is intrinsically prolonged by increasing the volume of the ethanol droplet. The increased immersion time leads to higher internal stress and more complete penetration of the liquid throughout the deeper levels of the MTF pore network, resulting in more profound structure distortion. Therefore, after longer immersion time, a higher number of MOF cells undergo conformational change and increase the total observed PL intensity.

It should be noted that interaction with the liquid ethanol is necessary for inducing the observed PL response, vapor sorption alone is insufficient. Liquid ethanol induces a significant structural perturbation in the MTFs that is not achievable via vapor sorption. This structural distortion is substantiated by the characteristic sloped plateaus (indicated by gray arrows) observed in overtone-normalized frequency ($\Delta f/n$) and half-bandwidth ($\Delta\Gamma/n$) (**Fig. 3** and Supplementary Fig. S25). In contrast, vapor sorption experiments do not exhibit these sloped plateaus (Supplementary Fig. S24). Therefore, the mechanism proceeds as follows: larger ethanol droplet volumes extend the

residence time of the liquid phase, thereby facilitating more profound structural distortion, which is evidenced by more pronounced drifts in the $\Delta f/n$ and $\Delta \Gamma/n$ (Supplementary Fig. S26). Consequently, a greater degree of structural distortion correlates with a higher PL intensity upon drying.

Furthermore, this PL enhancement phenomenon exhibits molecular size selectivity (**Figs. 2a and 2f**). Small methanol and ethanol molecules trigger a response, while larger isopropanol molecules do not. This size-dependence confirms that the intrinsic nanoporous nature of MTFs plays a role and the conformational change of ADC should be involved. Therefore, at the molecular level, this structural distortion is primarily attributed to the realignment of the ADC linkers. Molecular dynamics simulations further support this hypothesis as our results point towards a vertical rotation state of the central anthracene moiety of the ADC linkers during the transient drying stage. Our simulation results of single ethanol molecule passing through the MOF channels clearly indicate this movement as a possible cause for the rotation of the ADC linkers inside the MTF lattice (Supplementary Fig. 31a). Consequently, the continuous flux of ethanol molecules through the confined channels subjects the ADC rotors to a persistent re-aligned orientational state (**Fig. 4d**). Under this dynamic regime, the PL efficiency of the linker is significantly amplified. In this regard, the prolonged immersion time (associated with larger solvent volumes) facilitates structural distortion. The extended interaction between solvent molecules and MTF pores effectively increases the population of ADC linkers undergoing rotation towards a re-aligned state, thereby resulting in a proportional enhancement of the PL intensity.

We have refined the corresponding Fig.3 and Supplementary Figs. 24, 25, and 26 in the revised manuscript and Supplementary Information to clearly visualize these structural changes in the MOF layer. The figures referenced and text related in this response are also presented below for your review.

Page 8, line 29: “**Fig. 3** and Supplementary Fig. 25 illustrate the ethanol evaporation kinetics during the liquid immersion experiments. These data confirm that the immersion state itself does not yield an increase in PL. A significant PL enhancement is induced during the transient drying phase. The drying process spans ~1 minute and is characterized in the QCM data by an increase in frequency that is approximately linear with respect to time. **Notably, both the frequency and bandwidth signals exhibit a temporal drift when the layer is immersed (indicated by the gray arrows in Fig. 3).** These sloping plateaus reflect time-dependent changes in the properties of the MTFs during the immersion state, while a behavior is absent in the vapor sorption experiments (Supplementary Fig. 24). This contrast implies that liquid ethanol induces a structural distortion within the MOF layer that vapor exposure does not.”

Page 9, line 13: “As illustrated in Supplementary Fig. 26, the magnitude of PL enhancement increases with the volume of ethanol, a trend consistent with the results shown in Fig. 2e. QCM result reveals that larger droplet volumes prolong the immersion duration before the MTF dries. Extended exposure to the liquid ethanol correlates with a more pronounced drift in the QCM frequency and bandwidth signals, indicating more extensive structural perturbations within the film. Consequently, the stress generated during the transient drying stage increases or increasingly spreads across the whole MTF, resulting in a higher degree of PL enhancement.”

Fig. 1 Evolution of QCM and PL signals during ethanol evaporation. Top: Shifts in overtone-normalized resonance frequency, $\Delta f/n$, and half-bandwidth, $\Delta\Gamma/n$, upon dropping 20 μL ethanol on the quartz plate. The gray arrow points out the sloped plateaus in the QCM signal, indicating disturbance on MOF structure. Below: schematic diagram shows the different states during ethanol evaporation.

Supplementary Fig. 1 In-situ QCM and PL measurement acquired while sample is exposed to ethanol vapor. (a) Overtone-normalized frequency shifts (b) Overtone-normalized bandwidth shifts, and (c) Evolution of PL versus time. The artifacts from the vapor adding and removing are omitted in the PL signal.

Supplementary Fig. 2 QCM measurement. Shifts in overtone-normalized resonance frequency, $\Delta f/n$, and half-bandwidth, $\Delta\Gamma/n$, upon dropping 20 μL ethanol on the MOF thin film-coated resonator.

Supplementary Fig. 3. QCM and PL measurements with different volumes of liquid ethanol. (a1–a5) Shifts in overtone-normalized resonance frequency, $\Delta f/n$ (lower), and half-bandwidth, $\Delta\Gamma/n$ (upper), upon dropping 20 μL and 10 μL ethanol on the MOF thin film-coated resonator. For the overtone recorded, **a1**) 15 MHz, **a2**) 25 MHz, **a3**) 35 MHz, **a4**) 45 MHz, **a5**) 55 MHz. **(b)** PL measurement is performed in parallel.

Fig. 2 a, PL intensity (averaged from 410 to 430 nm) over time. The time region when the solvent was applied has been clipped, where an artefact was created by the glowing pipette tip not related to the MOF emission. Given in insets are photos taken of a sample in the three emission phases. These phases are “dry” (initial and final state), “immersion” (thin film solvent immersion on the substrate), and “light-up” (end phase of evaporation as solvent molecules leave the MOF pores). **f**, PL intensity measured during methanol and isopropanol evaporation. There is no luminescence turn-on for isopropanol.

Supplementary Fig. 4 (a) The process of a single ethanol molecule passes through MOF channel. A: Initial structure of ADC before the ethanol molecule enters. B: Upon entering the channel, the anthracene structure undergoes partial bending and rotation deformation. C: As the ethanol molecule is in the middle region of these four ADC molecules, the structures of ADC exhibit no significant deformation. D: The ethanol molecule is leaving the channel, the anthracene structure undergoes partial bending and rotation deformation again. E: As the molecule exits the channel, the anthracene structure reverts to its original form. For clarity, the Zn ions and DABCO molecules in the unit cell are omitted.

Fig. 3 d, The process of ethanol moves from both sides of ADC. Distance is the length of one ethanol molecule moving from bottom to top.

4. point 7 on Reflectivity Measurements. According to the authors, full reflectance spectra were impossible to collect due to the instrument limitation. While I understand that some characterization techniques may be challenging, reflectivity measurements are not among them, as they are routine in many laboratories. Given that the manuscript discusses optical film properties and they presented reflectivity data at different

wavelength (e.g., Figure 4c), such measurements are necessary to substantiate the claims. Their absence remains a significant gap.

Response: We thank the reviewer for this insightful comment. The diffuse reflectance spectrum of Zn-ADC-DABCO was acquired using a commercial spectrophotometer (Agilent Cary 7000). As shown in Supplementary Fig. 28, these data exhibit good agreement with the corresponding absorbance spectrum.

Given that the Agilent Cary 7000 spectrophotometer's internal setup precludes the acquisition of kinetic reflectance spectra during ethanol evaporation with a horizontally fixed sample, we employed another, bifurcated fiber-coupled measurement configuration utilizing a deuterium-halogen light source (Avalight-DH-CAL, Avantes) and a high-sensitivity spectrometer (AvaSpec-ULS-TEC-EVO, Avantes). One fiber serves as the illumination source while the second one collects the reflected signal. The probe was positioned in close proximity to the sample surface without intermediate optics ensuring signal collection at quasi-normal incidence. To mitigate scattering artifacts, the collected spectra were normalized against a standard Teflon reference and baseline-corrected at 852 nm. These data are presented in **Fig. R2**. The temporal evolution of the reflected signal was analyzed by integrating two distinct spectral bands (**Fig. R2a**): the 310–390 nm band, corresponding to the MTF's absorbing range, and the 650–730 nm non-absorbing window, where variations are primarily governed by solvent evaporation dynamics. In the 310–390 nm region, the signal exhibits a complex profile, a rise was observed upon ethanol addition. At ~105 and ~275 seconds, the reflectance shows a steady decline, followed by a sharp transient spike and decrease. Comparison with the non-absorbing baseline (650–730 nm) indicates these fluctuations to be significantly influenced by the ethanol droplet's changing curvature during the evaporation process. This suggests a complex case of specular (Fresnel) reflection—which depends on scattering and intrinsic absorption. Consequently, disentangling the variable scattering contribution (which is low in the solvated state but high in the dry state) from the absorption signal within the full reflectance spectrum is technically distinct. While the reflected signal shows a transient increase and decrease in the 320–340 s range (highlighted in blue in **Fig. R2a** and visualized spectrally resolved in **Fig. R2b**) likely correlating with the reorientation of the ADC linkers (the light-up state), distinguishing this structural signal from scattering artifacts remains ambiguous in this configuration. Therefore, we believe that the measurement employed in the main manuscript (**Fig. 4c**)—utilizing specific excitation and detection wavelengths at normal incidence coupled with optimized optical filtering—is superior for isolating the signal of interest. This targeted approach effectively suppresses scattering contributions, providing a more reliable time-resolved profile of the optical changes occurring during solvent evaporation.

We have clarified the result of steady-state diffuse reflectance spectrum in the revised manuscript (page 10, line 1):

“The steady-state reflectance spectrum of the Zn-ADC-DABCO thin film exhibits a pronounced dip in the 350–400 nm range (Supplementary Fig. 28), mirroring the features observed in its absorption spectrum.”

Supplementary Fig. 5 Reflectance characteristics of Zn-ADC-DABCO. Steady-state diffuse reflectance spectrum of the Zn-ADC-DABCO thin film.

Fig. R2 (a) Integrated reflected signal intensity (310–390 nm and 650–730 nm) versus time for the Zn-ADC-DABCO thin film during ethanol evaporation. 310–390 nm band corresponds to the absorbance range. In contrast, 650–730 nm region represents a non-absorbing window, where the variation in reflectance is primarily driven by ethanol evaporation. (b) Reflected signal spectra of the Zn-ADC-DABCO thin film recorded under deuterium-halogen lamp illumination. All spectra are normalized to a Teflon reference standard.

5. point 8 similar to point 4.

Response: As in our responses to comments #2 and #3, ethanol vapor sorption is insufficient to induce strong PL. The observed PL enhancement is driven by internal

stress generating by liquid ethanol and consequent structural realignment occurring during the transient drying stage.

We sincerely thank you for the constructive insights, which have significantly strengthened the quality of this work. We trust that the comprehensive revisions and additional reflectance data presented herein have fully addressed the concerns raised. We hope that the revised manuscript is now considered suitable for publication in Nature Communications.

RESPONSE TO REVIERS' COMMENTS

Reviewer #3 (Remarks to the Author):

I thank the authors for their response and their efforts in improving this work. Considering all their answers and modifications, I think that the manuscript can be published.

Response: We thank you for your appreciation of our work and the constructive suggestions given.